# Evaluating cloud properties in an ensemble of regional on-line coupled models against satellite observations

Rocío Baró[1,2], Pedro Jiménez-Guerrero[1], Martin Stengel[3], Dominik Brunner[4], Gabriele Curci[5,6], Renate Forkel[7], Lucy Neal[8], Laura Palacios-Peña[1], Nicholas Savage[8], Martijn Schaap[9], Paolo Tuccella[5,6], Hugo Denier van der Gon[9], and Stefano Galmarini[10]

[1]Department of Physics, Regional Campus of International Excellence Campus Mare Nostrum, University of Murcia, Murcia, Spain
[2]Now at: Section Chemical Weather Forecasts, Division Data/Methods/Modelling, ZAMG – Zentralanstalt für Meteorologie und Geodynamik, Vienna, Austria
[3]Deutscher Wetterdienst (DWD) Frankfurter Str. 135 Offenbach, Germany
[4]Laboratory for Air Pollution and Environmental Technology, Empa, Dübendorf, Switzerland
[5]Department of Physical and Chemical Sciences, University L'Aquila, L'Aquila, Italy
[6]Center of Excellence in Telesening of Environment and Model Prediction of Severe Events (CETEMPS), University of L'Aquila, L'Aquila (AQ), Italy
[7]Karlsruher Institut für Technologie (KIT), Institut für Meteorologie und Klimaforschung, Atmosphärische Umweltforschung (IMK-IFU), Germany
[8]Met Office, FitzRoy Road, Exeter EX1 3PB, UK
[9]Netherlands Organization for Applied Scientific Research (TNO), Utrecht, The Netherlands
[10]European Commission, Joint Research Centre (JRC), Directorate for Energy, Transport and Climate, Air and Climate Unit, Ispra (VA), Italy

*Correspondence to:* Pedro Jiménez-Guerrero (pedro.jimenezguerrero@um.es)

**Abstract.**

On-line coupled meteorology-chemistry models permit the description of the aerosol-radiation (ARI) and aerosol-cloud interactions (ACI). The aim of this work is to assess the representation of several cloud properties in regional–scale coupled models when simulating the climate-chemistry-cloud-radiation system. The evaluated simulations are performed under the umbrella of the Air Quality Model Evaluation International Initiative (AQMEII) Phase 2 and include ARI+ACI interactions. Model simulations are evaluated against observational data from the European Space Agency (ESA) Cloud_cci project. The results show an underestimation (overestimation) of cloud fraction (CF) over land (sea) areas by the models. Lower bias values are found in the ensemble mean. Cloud optical depth (COD) and cloud ice water path (IWP) are generally underestimated over the whole European domain. The cloud liquid water path (LWP) is broadly overestimated. The temporal correlation suggests a general positive correlation between models and satellite observations. Finally, CF gives the best spatial variability representation, whereas COD, IWP and LWP show less capacity. The differences found can be attributed to differences in the used microphysics schemes; for instance, the number of ice hydrometeors and the prognostic/diagnostic treatment of the LWP are relevant.

# 1 Introduction

Atmospheric aerosols vary in time and space, influence the Earth's radiation budget and can lead to variations in cloud micro-physics, which impact cloud radiative properties and climate. These processes have traditionally been called the aerosol direct effect, but were renamed after the Fifth Report of the Intergovernmental Panel on Climate Change (IPCC AR5) (Boucher et al., 2013; Myhre et al., 2013) as aerosol-radiation interactions (ARI). Furthermore, aerosols serve as cloud condensation nuclei (CCN) that influence overall cloud radiative properties through interactions referred to as the first indirect effect or Twomey effect (Twomey, 1974, 1977). More aerosol particles lead to more cloud condensation nuclei, which results in an increased concentration of cloud droplets. When the cloud water is fixed, it is accompanied by a reduced cloud droplet size and increased cloud reflectivity. Altogether this results in less solar energy absorbed and a cooling of the system. Aerosols that act as CCN may affect precipitation efficiency, cloud life–time, and cloud thickness, and could thus further influence weather and climate through the second indirect effect (Albrecht, 1989), also namely the cloud lifetime effect. The modification of cloud micro-physical properties is expected to have an impact on the cloud evolution, particularly in terms of cloud's ability to generate large enough droplets to initiate precipitation. This effect is traditionally called the second aerosol indirect effect, but since the AR5, these indirect effects are called aerosol–cloud interactions (ACI). Those interactions are more uncertain due to the complexity of the microphysical processes (Boucher and Lohmann, 1995; Schwartz et al., 2002; Lohmann and Feichter, 2005).

The inclusion of aerosol interactions in air quality/climate modelling is an important challenge, and is also important for the development of integrated emissions control strategies for both air quality management and climate change mitigation (Yu et al., 2014; Rosenfeld et al., 2014). There are different approaches to address the study of ACI, usually by combining methodologies of observations and/or modelling. In the field of observations/remote sensing, McComiskey et al. (2009) used the Atmospheric Radiation Measurement (ARM) focused on the California area. These authors studied the albedo effect as the change in cloud droplet number concentration (CDNC) with aerosol concentration, which resulted in local radiative forcing of around $-13$ W m$^{-2}$ (top-of-the-atmosphere). Liu et al. (2011) also used ARM combined with GOES satellite measurements, and theoretically derived an analytical relationship, linking relative surface shortwave cloud radiative forcing, cloud fraction and cloud albedo. They noticed its utility for diagnosing deficiencies of cloud-radiation parameterisations in climate models. By using observations and modelling, Avey et al. (2007) employed cloud retrievals from the Moderate Resolution Imaging Spectroradiometer (MODIS) and output from a tracer transport model (FLEXPART). They compared cloud and pollution fields on the northeastern coast of the United States, during 2004, under the umbrella of International Consortium for Atmospheric Research on Transport and Transformation (ICARTT) mission. They found that, where the transport model indicated polluted air, cloud droplet effective radii was smaller, while cloud optical depth (COD) was greater in some cases, or at least close to primary source regions. They found no conclusive evidence for the perturbation of the cloud liquid water path (LWP) by pollution. Yang et al. (2011) used the Weather Research Forecast coupled with a chemistry (WRF-Chem) model in a study conducted over the northern Chilean and southern Peruvian coasts from 15 October to 16 November 2008. They ran a simulation including ACI and compared to other run with fixed CDNC and simplified cloud and aerosol treatments. They found that the coupled simulation of ACI improved cloud optical and microphysical properties.

In order to realistically simulate the chemistry-aerosol-cloud-radiation-climate interactions, fully online-coupled meteorology-atmospheric chemistry models are needed (Baklanov et al., 2008; Zhang, 2008). Moreover, to build confidence in air quality-climate interaction studies, a thorough evaluation is needed on both global and regional scales. Particularly, ACI is still considered one of the most important uncertainties in anthropogenic climate perturbations (Penner et al., 2006; Quaas et al., 2009).

The air quality model evaluation international initiative (AQMEII) (Rao et al., 2011) was set up to promote research into regional air quality model evaluations across the regional modelling communities in Europe and North America. This study is conducted in the context of Phase 2 of AQMEII where model evaluation is made in on-line-coupled air quality models. An extensive model evaluation of the simulations showed herein can be found in Brunner et al. (2015) and in Im et al. (2015a, b). There is a follow-up of the AQMEII intiative, Phase 3, which focuses on evaluating and intercomparing regional and linked

global/regional modelling systems by collaborating with the Task Force on Hemispheric Transport of Air Pollution, Phase 2 (Janssens-Maenhout et al., 2015).

To the author's knowledge, apart from the study of Makar et al. (2015a), there are no other studies that have taken into account ARI+ACI in regional coupled models. The main objective of this contribution is to assess the representation of several cloud variables in different regional–scale integrated models when simulating ARI+ACI. To date, all the collectives studies

performed used global models and regional climate analyses do not usually bear in mind ARI+ACI. In the next section, we explain the methodology followed, where we provide an overview of the model simulations, the description of the observational data used and the evaluation methodology. In Section 3, the results of the evaluation of the assessed cloud properties and the spatial correlation and variability are described. The paper closes with a summary and conclusions.

## 2 Methodology

This Section describes the strategies adopted to analyze cloud properties in online-coupled models. As stated in the ntroduction, the analyzed model outputs are the results run according to the AQMEII Phase 2 initiative. In order to analyze the capacities of the coupled models which take into account ARI+ACI, simulations from different models with identical meteorological boundary conditions and anthropogenic emissions have been analyzed.

The common set-up for the participating models and a unified output strategy allowed us to analyse the representation of

25 model output in relation to similarities and differences in the model's response to the aerosol-radiation and aerosol-cloud interactions. The studied variables are the cloud fraction (CF), the cloud optical depth (COD), the cloud ice path (IWP) and the cloud water path (LWP). The target domain is Europe, and the analysis covers the year 2010 and its seasonality.

### 2.1 Model simulations

Table 1 offers an overview of the one-year model simulations that contribute to this study in the AQMEII Phase 2 context. It

includes five simulations conducted with the following online-coupled models: LOTOS-EUROS (Sauter et al., 2012), UKCA (Savage et al., 2013) and WRF-Chem (Grell et al., 2005; Grell and Baklanov, 2011). LOTOS-EUROS is a semi-online model where the two models run separately, but wait for one another to exchange information (meteorology and aerosol concentra-

tions) every 3 hours. Cloud fields were an optional part of the variables to be submitted and not all models were able to provide all these fields within the limited time available. Of the 13 models in AQMEII 2 which modelled the European domain, only five provided any of these fields, and only three presented a complete set. To facilitate the cross-comparison between models, the participating groups interpolated their model output to a common grid at the 0.25° resolution (except for NL2 model, which had a smaller grid).

All the simulations were driven by the European Centre for Medium-Range Weather Forecasts (ECMWF) operational analyses (with data at 00 and 12 UTC) and with respective forecasts (at 3/6/9, etc., hours), so that the time interval of meteorological fields used for the boundary conditions was 3 hourly. The chemical initial conditions (IC) were provided by the ECMWF IFS-MOZART model. The employed anthropogenic emissions were provided by the Netherlands Organization for Applied Scientific Research (TNO). The dataset is a follow-on to the widely used TNO-MACC database (Pouliot et al., 2012). Biogenic emissions were estimated by the Model of Emissions of Gases and Aerosols from Nature (MEGAN) (Guenther et al., 2006), which were calculated online. Fire emissions data were obtained from the IS4FIRE Project (http://is4fires.fmi.fi). The emission dataset is estimated by a re-analysis of the fire radiative power data obtained by the MODIS instrument onboard the Aqua and Terra satellites. For further information about the models' parameterisations, the reader is referred to Brunner et al. (2015); Im et al. (2015a, b).

## 2.2 Observational data

In order to analyze the representation of the different cloud properties, model data were compared and evaluated against the satellite-based observations of cloud properties. In more detail, the satellite data were generated by the ESA Cloud_cci project, within the ESA's Climate Change Initiative (CCI) programme (see Hollmann et al., 2013, for scientific aspects covered in the CCI programme). Several datasets are generated in Cloud_cci (Stengel et al., 2017a), and in this study, the Level-3C data (monthly averages and histograms) of the Cloud_cci AVHRR-PM dataset (Stengel et al., 2017b) were used. Data were retrieved by employing the Community Cloud retrieval for Climate (CC4CL, Sus et al., 2018; McGarragh et al., 2018) using measurements of the Advanced Very High Resolution Radiometer (AVHRR) onboard the National Oceanic and Atmospheric Administration satellite No. 19 (NOAA-19). CC4CL itself consists of three parts: cloud detection, cloud phase assignment and the retrieval of cloud properties (e.g. optical thickness and effective radius). For the latter, scattering properties of liquid clouds are determined following Mie theory code as implemented by Grainger et al. (2004) using a modified gamma distribution to which the effective radius, which parametrizes the size distribution, is related to. For ice clouds, the ice crystal single-scattering models of Baum et al. (2011, 2014) are used, with the bulk single-scattering properties being determined by an integration over particle size distribution of nine ice particle habits. For more information see McGarragh et al. (2018).

The Level 3C data were used in this study, which has a spatial resolution of 0.5° Latitude/Longitude and represents a monthly mean of instantaneous cloud property retrievals taken at 01:30 AM/PM local time. The dataset version used was v2.2, which contained, compared to Stengel et al. (2017a) who described v2.0, two significant bug fixes: (1) correcting a miscalculation of bidirectional reflectance distribution function (BRDF) components under the condition of high solar zenith angles and/or snow/ice covered surfaces; (2) correcting look-up tables with pre-calculated radiances according to ice cloud properties, as well

as viewing geometry and illumination condition. Both bug fixes lead to a significant reduction in the random and systematic uncertainties of the data, particularly for the optical properties cloud effective radius and cloud optical thickness, as well as those from the derived cloud liquid and ice water path.

In preparation for the presented study, the cloud mask validation presented in Stengel et al. (2017a), was redone but limited to the European area, which showes biases of approximately $-13\%$ in Cloud_cci. After removing all clouds with optical thicknesses below 0.15, the biases are nearly vanished. Separating land and ocean regions did not indicate any significant difference in cloud detection efficiency between these two surface types for the European area. In addition, Cloud_cci IWP was validated against DARDAR (raDAR/liDAR cloud parameter retrievals) products (Delanoë and Hogan, 2008, 2010). For global collocations, the Cloud_cci bias amounts to $-114$ g m$^{-2}$ compared to DARDAR. Most significant underestimations of IWP occur for large IWPs (above 500 g m$^{-2}$).

## 2.3 Evaluation methodology

Regarding the model evaluation methodology, satellite data are bilinearly interpolated to a common working grid covering the European domain. For the evaluation of cloud variables, model data were postprocessed by computing the monthly mean of the mean value from 13.00 to 14.00 PM. In order to evaluate the studied variables, several classical statistics were used according to Willmott et al. (1985) and Weil et al. (1992). We computed the mean bias error (bias) and the correlation coefficient. The computation of the median showed identical spatial patterns, so only the mean results are shown.

The bias (Eq. 1) is defined as:

$$Bias = \frac{1}{n} \sum_{i=1}^{n} e_i = \bar{P}_i - \bar{O}_i \tag{1}$$

Where $e_i$ is the individual model-prediction errors usually defined as prediction ($P_i$) minus observations ($O_i$), and $\bar{P}$ and $\bar{O}$ are the model-predicted and observed means, respectively.

The standard deviation of the $P_i$ (Eq. 2) is:

$$\sigma_P = \sqrt{\frac{1}{n} \sum_{i=1}^{n} (P_i - \bar{P})^2} \tag{2}$$

The standard deviation of the $O_i$ (Eq. 3) is:

$$\sigma_O = \sqrt{\frac{1}{n} \sum_{i=1}^{n} (O_i - \bar{O})^2} \tag{3}$$

The correlation (Eq. 4) is:

$$r = \left[ \frac{\frac{1}{N}\sum_{i=1}^{n}(O_i - \bar{O})(P_i - \bar{P})}{\sigma_O \sigma_P} \right] \tag{4}$$

The standard deviation ratio was computed as $\sigma_p/\sigma_o$. A satellite data mask for each monthly mean was done and applied to model data in order to compute the statistics over the same area. The mean values were computed, and are discussed in Section 3. Since satellite data availability was monthly means, the temporal coefficient of correlation is only shown for the whole year (2010). To compute the correlation, a satellite data mask containing more than or equal to 6 months of satellite data was considered so that, the correlation is shown only in the grid points where there are at least 6 months of data.

## 3 Results

This Section describes the behaviour of the studied variables (CF, IWP, LWP, COD) for the bias, temporal correlation and spatial variability. They were obtained by calculating the corresponding statistic of the monthly mean series at each grid point of all the land grid points of the domain for each season as follows: January-February-March (JFM); April-May-June (AMJ); July-August-September (JAS); October-November-December (OND). The continental European domain includes the North of Africa, western part of Russia and Iceland. All the figures in the present study have the same structure (but temporal correlation): top row represents the mean satellite values from ESA Cloud_cci (discontinuity features seen around 60N are due to small inconsistencies between day, twilight and night-time retrievals, which are found most prominent in regions with day and night-time observations border on regions without any day-time observations). From left to right, the mean for the analyzed periods: 2010, JFM, AMJ, JAS and OND are depicted. The following rows (2 to 7) include the computed statistic for each model and the ensemble (ENS) mean, estimated as the average of all the available model simulations. The yearly correlation is shown for the temporal correlation. The first row shows the mean satellite data for the cloud variables (one in each column) for 2010, while the following rows show the temporal correlation for each simulation.

### 3.1 Cloud fraction, CF

Fig. 1 shows the bias for the variable CF. In both cases, the first row shows the satellite CCI values, which are generally higher than 0, with minimum values over the eastern Mediterranean that increase with latitude. The values between 0 and 1% are found in some areas during summer months, mainly in northern Africa. The following rows show the bias of the different model simulations. Table 2 provides the mean values of satellite, models and ENS. For CF, the mean model values come close to the satellite data, with a slight tendency for underestimation. Fig 1 generally indicates an underprediction for CF over land areas and an overestimation over the ocean. Individual model simulations present a bias range from +40% to over −35% over the studied region. The ES1 model presents the highest underestimation (−40% mean bias), mainly over land areas. For the ENS mean (last row in Fig 1), lower values are found, with biases ranging from 20% to −20%, which outperform the individual simulations. The positive bias is more marked during JAS, where the mean satellite values are lower (first row, in Fig 1). A

negative bias is expected because of the general trend of global and regional models to underestimate CCN (Wyant et al., 2015) and, therefore, cloud formation. The overestimations found off-shore could be produced because satellite retrievals missed thin clouds. Lastly, Fig. 2 represents the mean satellite data for the cloud variables in the first row for 2010. The following rows cover the temporal correlation for each model simulation. For CF, a positive temporal correlation prevails, with mean values of

0.7/0.8 and areas with values that come close to 0.9. Conversely, there are some areas over the sea, with a negative correlation (around $-0.5$). This spatial pattern of the correlation coefficient is related to Fig. 1, where a negative bias prevails over land areas and a positive one over sea. Generally, a positive correlation implies that when the satellite CF values increase (decrease), the model's CF values increase (decrease) but models underestimate this mainly over land areas (Fig. 1).

### 3.2 Cloud ice path, IWP

Regarding IWP, all-sky mean was computed (also for the LWP variable). Fig. 3 presents the mean satellite values (first row), which are below 100 g m$^{-2}$, but for some delimited areas during winter months (JFM, OND) and spring (AMJ), where higher values are found (over 200 g m$^{-2}$). The third column in Table 2 reflects that the mean models values are significantly lower compared to the satellite retrievals for IWP. Therefore, the IWP bias in Fig. 3 shows a general model underestimation, but for UK4. The WRF-Chem models (DE4, ES1, IT2) show negative biases between $-80$ and $-50$ g m$^{-2}$ in different parts of

the domain, depending on the season. The largest underestimations are found during JFM and OND (where the mean satellite values are very high). On the other hand, UK4 overestimates the IWP, with a positive bias of 80 g m$^{-2}$ during JFM over central and northern Europe. During JFM, the mean satellite data were around 50 g m$^{-2}$, which is best captured by the other models. UK4 also overestimates IWP for the rest of the year over some northern areas of the domain. The differences found here in relation to WRF-Chem models and UK4 could be related to the number of hydrometeors defined in each microphysics scheme.

For both WRF-Chem microphysics (Lin et al., 1983; Morrison et al., 2009), three types of ice hydrometeors are considered, whereas UK4 considers only one (Wilson and Ballard, 1999). IWP is a prognosticked variable in UK4 model. The fact that the WRF-Chem simulations underestimates the IWP, with an overestimation in the UK4 model, could mean that the number of ice hydrometeors in the microphysics scheme is relevant for the IWP representation. At the same time, the ENS simulation outperforms the individual simulations because it compensates the UK4 model overestimation with the underestimations of the

other models. The temporal correlation (Fig. 2) shows positive correlation values around 0.7 and negative correlations between $-0.5$ and $-0.6$. Positive correlations are practically found in the entire domain, whereas negative correlations are found in northern Europe (Scandinavian area and north of Russia). Since the mean models values are significantly lower compared to the satellite retrievals and, together with the negative CCI bias against DARDAR, strengthens the conclusion that models have a very small IWP.

### 3.3 Cloud water path, LWP

The bias of the LWP is shown in Fig. 4. The mean satellite values (first row, Figs. 4) are below 100 g m$^{-2}$ (as well as for IWP) but values are higher than 150 g m$^{-2}$ in winter months (during JFM, mainly in the North of Spain, some areas of the Mediterranean coast, France and North and the Baltic Sea and in OND). As for the IWP during OND, the LWP is higher over

the entire domain (except for North Africa). Greenwald et al. (1993) used the special sensor microwave/imager (SSM/I) to retrieve integrated LWP, which found values around 100 g m$^{-2}$. Although the mean satellite data seems to be in agreement with other studies, the models shows higher LWP values. When the mean model values,except for the ES1 model (last column, Table 2) are higher compared to satellite values, we can see in Fig. 4 a general overestimation of the LWP (values up to +50

5   g m$^{-2}$) and mainly over the sea. The model differences found here could be related to the LWP treatment in the model. For instance, in all the models except UK4, the LWP is treated as an prognostic variable whereas UK4 treats it as a diagnostic variable (Wilson and Ballard, 1999). Besides, within the WRF-Chem models and according to Baró et al. (2015), the models with a Morrison scheme have more droplets with a smaller diameter compared to the Lin scheme. This could also affect the representation of this variable, where the ES1 model underestimates the LWP over most of the domain. According to Tiedtke

(1993), a correct representation of the LWP is important for high clouds, because it is directly related to the transparency or optically thickness. As it will be shown in Section 3.4, NL2 underestimates COD, (explained by the fundings of Tompkins et al. (2007) when testing the scheme). No data are available to evaluate the IWP or LWP over northern Africa. The temporal correlation (Fig. 2) shows a positive correlation value of around 0.7 for most of the domain. Negative correlations prevail in the Atlantic Ocean and some parts in central Europe (up to −0.6).

**3.4   Cloud optical depth, COD**

Regarding COD, the mean seasonal satellite values (first row in Fig 5) go up to 30, with the highest mean values during the OND period. The lowest values are found over northeastern part of the domain, with values under 10. The second column of Table 2 second column indicates that generally lower mean model values are found compared to the satellite data. This spatial pattern is clear in the COD bias (Fig. 5), with a general underestimation of the monthly mean COD over the whole domain.

In general, a higher negative bias is found during OND, and NL2 gives the largest underestimation (values up to −30). In winter months (JFM), DE4 and IT2 show an overestimation over central Europe and some areas around +15, over the Atlantic Ocean, which coincide with the low COD values in the satellite. For the WRF-Chem models, the differences that appear between models can be related to the different microphysics scheme (Table 1) employed; Morrison (Morrison et al., 2009) in DE4 and IT2 and the Lin scheme (Lin et al., 1983) in ES1. According to Baró et al. (2015), which studied the differences

between these microphysics schemes, Morrison parameterisation involves higher droplet number mixing ratio values. The authors stated that, since cloud water is similar for the Morrison and Lin simulations, the higher droplet number mixing ratio in Morrison indicates that cloud droplets have a smallers diameter in Morrison than in Lin (especially during winter). Since COD measures the attenuation of radiation due to extinction by cloud droplets, smaller and more cloud droplets in the Morrison scheme are more effective in scattering shortwave radiation, and could explain the positive biases found in DE4 and

IT2 models. The differences found in the NL2 model may once again be related to its model microphysics scheme (Table 1)(Tiedtke, 1993; Tompkins et al., 2007; Neggers, 2009). Tompkins et al. (2007) tested the new scheme in the European Centre for Medium-Range Weather Forecasts (ECMWF), Integrated Forecasting System (IFS) model within two 7-member ensembles of 13-months. They made a comparison with the ISCCP D2 retrievals, and found a reduction of the high-cloud cover leading to a lower COD. This coincides with the results found herein, where this model presents the largest underestimations. Once

again, the ENS simulation outperforms the individual simulations. Regarding the temporal correlation (second column in Fig. 2), a general positive correlation is seen with values up to 0.8 (mostly over land areas) and others with a negative correlation over central Europe in models DE4 and IT2, which coincides with those areas where the bias is overestimated.

### 3.5 Spatial correlation and variability

The spatial correlation and variability, averaged for year and target season, are summarised in Table 3 for each variable (CF, COD, IWP and LWP, and in that order). For the CF, the seasonal correlation coefficients are very high (over 0.90 for each model and season, except for ES1 in wintertime, with a correlation = 0.89). Yearly correlation coefficients range from 0.85 to 0.89, which indicates that the models are well able to capture the spatial variability of the CF. The $\sigma_P/\sigma_O$ ratio provides an idea of the trend of the simulations to overestimate or underestimate the spatial variability (ratio over or under 1, respectively).

All the models present accurate spatial variability representativity, with ratios coming very close to 1 for every season, and also for the annual average. All the models have a very slight tendency to overestimate CF spatial variability (the $\sigma_P/\sigma_O$ ratio ranging from 1.01 to 1.07).

The spatial correlation coefficient for the other variables indicates less capacity to represent the spatial correlation of COD, IWP and LWP. All the annual spatial correlation values are in the order of 0.6-0.7, and range from the case of NL2 for
COD (0.41) and LWP (0.44) at the bottom to the simulation of UK4 for the LWP (0.73). These values are similar if seasonal correlation coefficients are observed, except for summertime and the IWP variable. The model's capacity to represent the IWP spatial pattern is limited during JAS, with correlation coefficients ranging from 0.16 in UK4 to 0.35 in DE4 and IT2. Once again, the Morrison microphysics seems to outperform all the other simulations when representing the cloud ice path.

As for the spatial variability of COD, IWP and LWP, represented by the $\sigma_P/\sigma_O$ ratio, major differences between the variables
and models are found. For the COD, ES1 and NL2 tend to underestimate its spatial variability, especially for NL2, with $\sigma_P/\sigma_O$ values ranging from 0.09 during OND to 0.17 for summertime (JAS). The other models present a good capacity to reproduce variability, with slightly higher ratios for the yearly-averaged values than for individual seasons. For the IWP, spatial variability is generally estimated by all the models and seasons ($\sigma_P/\sigma_O$ values in the order of 0.1-0.2), except for UK4, which slightly underpredicts variability (ratios around 0.8, except for OND, when this value drops to 0.63).

Lastly, for the LWP, all the models but ES1, slightly overestimate the spatial variability ($\sigma_P/\sigma_O$ values around 1.0-1.3) for the yearly-averaged values in winter and spring. In summer, this value is slightly overestimated by the simulations that do not use WRF-Chem (around 1.4 for NL2 and 1.2 for UK4), while for autumn (OND) all the models tend to underpredict the spatial variability (values of 0.7/0.8). In general, the best capacities are found for the DE4 simulations, while the largest underestimations are present in the ES1 simulations with values around 0.6, which use the Lin microphysics scheme.

### 4 Summary and conclusions

The presence or the absence of cloudiness must be well represented in modelling, since clouds play an important role in the Earth's energy balance (Boucher et al., 2013; Myhre et al., 2013). Hence a collective evaluation of the cloud variables CF,

COD, IWP and LWP is shown in this study. The simulations evaluated herein were run by coupled chemistry and meteorology models in the AQMEII Phase 2 initiative context for the year 2010. This study complements other collective analyses, such as Baró et al. (2017); Brunner et al. (2015); Makar et al. (2015a, b) and Forkel et al. (2015) by adding an assessment of how online-coupled models represent some cloud properties in an ensemble of simulations.

As for the CF, an underestimation (overestimation) of this variable is observed over land (sea) areas. Individual model simulations present a positive bias close to 40% and a negative bias over $-35\%$. For the mean ENS, lower CF values are found, with biases ranging from 20% to $-20\%$, which outperform individual simulations. The positive bias is more pronounced during JAS, where the mean satellite values are lower. The negative bias may be due to the general underestimation in the CCN representation by global and regional models (Wyant et al., 2015). The overestimations found off-shore might be related

to satellite retrievals missing thin clouds. A positive temporal coefficient of correlation dominates in the spatial pattern of CF, (values close to 0.9) and a negative correlation (around $-0.5$) in some areas over the sea. This is similar to the bias, where a negative bias prevailed over land areas and a positive bias over the sea.

There is an overall underestimation of the IWP, except for UK4. The differences found here in relation to the WRF-Chem models and UK4 could be related to the number of hydrometeors defined in each microphysics scheme. For both WRF-

Chem microphysics, three types of ice hydrometeors are considered, whereas UK4 (Wilson and Ballard, 1999) considers only one. So the overestimation found in the UK4 model could mean that the number of ice hydrometeors is relevant. The temporal correlation shows positive correlation values at around 0.7 and negative correlations between $-0.5$ and $-0.6$. A positive correlation is found over nearly the whole target domain, whereas negatives correlations are found in northern Europe (Scandinavian countries and the north of Russia).

Despite the LWP mean satellite data seeming to be in agreement with other studies (Kniffka et al., 2014), models shows higher LWP values that result in a general LWP overestimation, mainly over sea areas (except the ES1 model). The model differences found here could be related to the treatment of the variable because, for instance, in all the models except for UK4, LWP is treated as an prognostic variable, whereas UK4 treats it as a diagnostic variable (Wilson and Ballard, 1999). As seen in Section 3.4, in the WRF-Chem models and according to Baró et al. (2015), the models with the Morrison scheme have

more droplets with a smaller diameter compared to the Lin scheme. This could also affect the representation of this variable by showing an ES1 model underestimation over most of the domain. According to Tiedtke (1993), a correct representation of the LWP is important for high clouds, given its directly relation to the transparency or optically thickness. Besides, as mentioned in Section 3.4, NL2 underestimates COD. The temporal correlation shows positive values at around 0.7 for most of the domain. Negative correlations prevail in the Atlantic Ocean and some parts of central Europe (up to $-0.6$).

Regarding COD, the lower mean model values are found compared to the satellite data resulting in a general underestimation of the monthly mean over the whole domain. A generally higher negative bias is found during OND, with NL2 showing the largest underestimation. In winter, DE4 and IT2 trend to overestimate over central Europe and some areas over the Atlantic Ocean, which corresponds to low COD values, as indicated by the satellite. These differences in the WRF-Chem models may be related to the different microphysics scheme used (Morrison (Morrison et al., 2009) versus Lin (Lin et al., 1983)). In the

former, cloud droplets have a smaller diameter than Lin (especially during winter) (Baró et al., 2015), which leads to more

effective extinction by cloud droplets. The differences found in the NL2 model may be related to the model microphysics scheme. Temporal correlation indicates a general positive correlation between models and satellite observations, with values up to 0.8 (mostly over land areas). Some areas with a negative correlation over central Europe in models DE4 and IT2 are related to areas with an overestimation trend.

Finally, the seasonal and yearly correlation coefficients are very high for the CF (seasonal over 0.90, yearly over 0.85), which indicates that the models are well able to capture the spatial variability, while they tend to slightly overestimate CF spatial variability ($\sigma_P/\sigma_O$ values ranging from 1.01 to 1.07). The other variables indicate less capacity to represent the spatial correlation of the COD, IWP and LWP. All the annual spatial correlation values are in the order of 0.6-0.7, which are similar when seasonal correlation coefficients are observed, except for the IWP in the summertime. The model's capacity to represent

the IWP spatial pattern is limited during JAS, (correlation coefficients ranging from 0.16 to 0.35). Morrison microphysics seems to outperform the other simulations when representing the cloud ice path. Major differences in the spatial variability between the variables and models are found. For COD, ES1 and NL2 tend to underestimate its spatial variability, especially for NL2, with $\sigma_P/\sigma_O$ values ranging from 0.09 during OND to 0.17 for the summertime (JAS). The other models present a good capacity to reproduce the variability, with slightly higher ratios for the yearly-averaged values. With the IWP, the

spatial variability is pervasively estimated by all the models and seasons, except for UK4, which slightly underpredicts the variability (with ratios around 0.8, except for OND, when this value drops to 0.63). For the LWP, all the models but ES1 slightly overestimate the spatial variability ($\sigma_P/\sigma_O$ values around 1.0-1.3) for the yearly-averaged values, winter and spring. In summer, the best capacities go to the DE4 simulations, while the largest underestimations are present for ES1 simulations (which use Lin microphysics scheme).

According to Rosenfeld et al. (2014), a better understanding of the aerosol-cloud processes would reduce the uncertainty in anthropogenic climate forcing and provide a clear understanding and better predictions of the future impacts of aerosols on both climate and weather. With this study, it has been shown, how the online coupled models represent several cloud properties, which complements the temperature collective analyses of Baró et al. (2017).

*Acknowledgements.* Special acknowledgment to the ESA CCI Cloud team who provided us the Cloud data for doing this study. We also

acknowledge the initiative AQMEII2 and the Joint Research Center Ispra and the Institute for Environment and Sustainability for its EN-SEMBLE system. The group from University of L'Aquila kindly thanks the EuroMediterranean Centre on Climate Change (CMCC) for the computational resources. P.T. is beneficiary of an AXA Research Fund postdoctoral grant. The Project REPAIR (CGL2014-59677-R), funded by the Spanish Ministerio de Economía y Competitividad (MINECO) and the FEDER European program has supported the accomplishment of this study. The author Rocío Baró acknowledges the FPU scholarship (Ref. FPU12/05642) by the Spanish Ministerio de Educación, Cul-

tura y Deporte. Pedro Jiménez-Guerrero acknowledges the fellowship 19677/EE/14 of the Programme Jiménez de la Espada de Movilidad, Cooperación e Internacionalización (Fundación Séneca-Agencia de Ciencia y Tecnología de la Región de Murcia, PCTIRM 2011-2014).

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

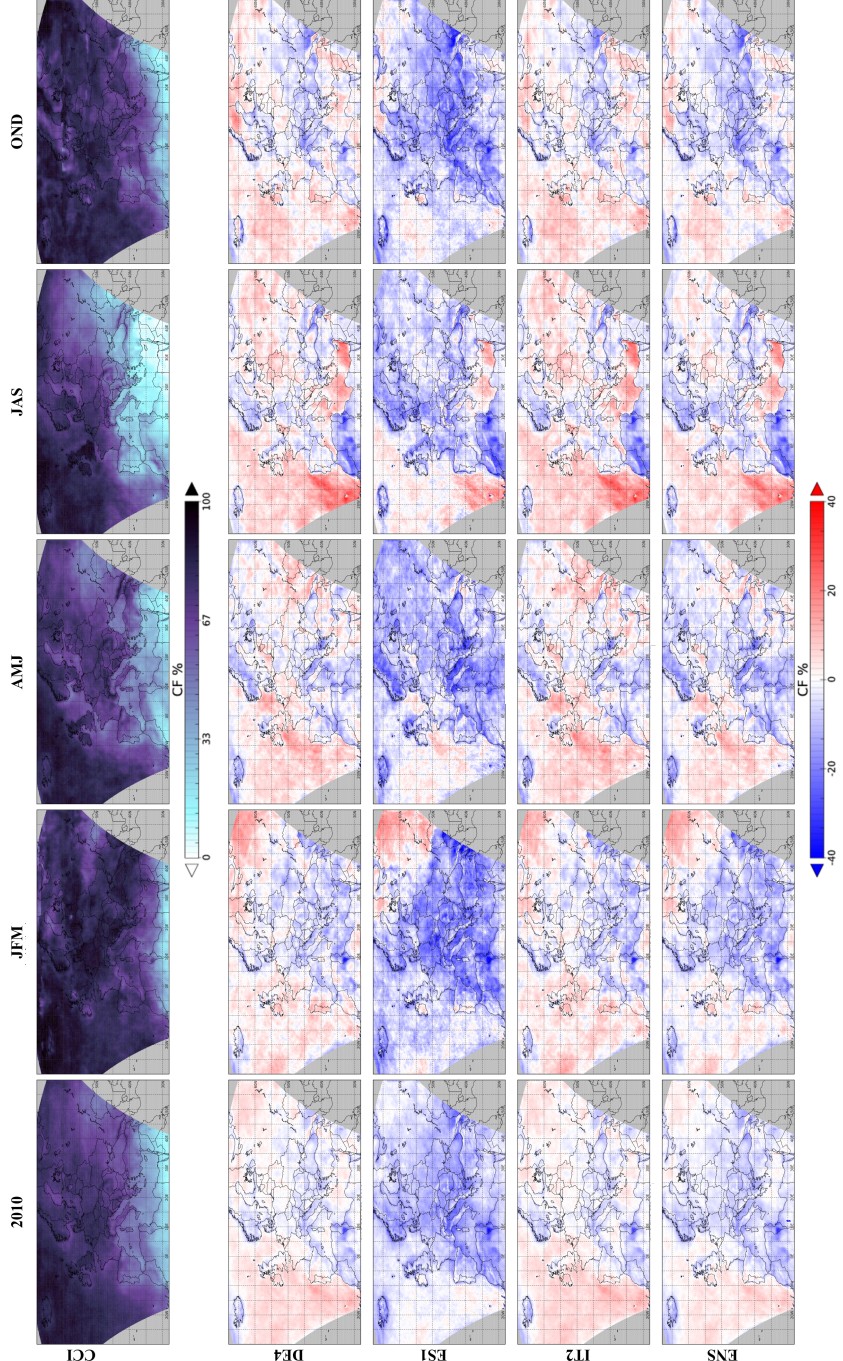

**Figure 1.** MEAN BIAS ERROR (bias) CF. First row represents the mean satellite values of 2010 (first column), JFM (second column), AMJ (third column), JAS (fourth column), OND (fifth column). Following rows represent the bias of the models: DE4 second row, ES1 third row, IT2 fourth row, NL2 fifth row, UK4 sixth row and ENS seventh row.

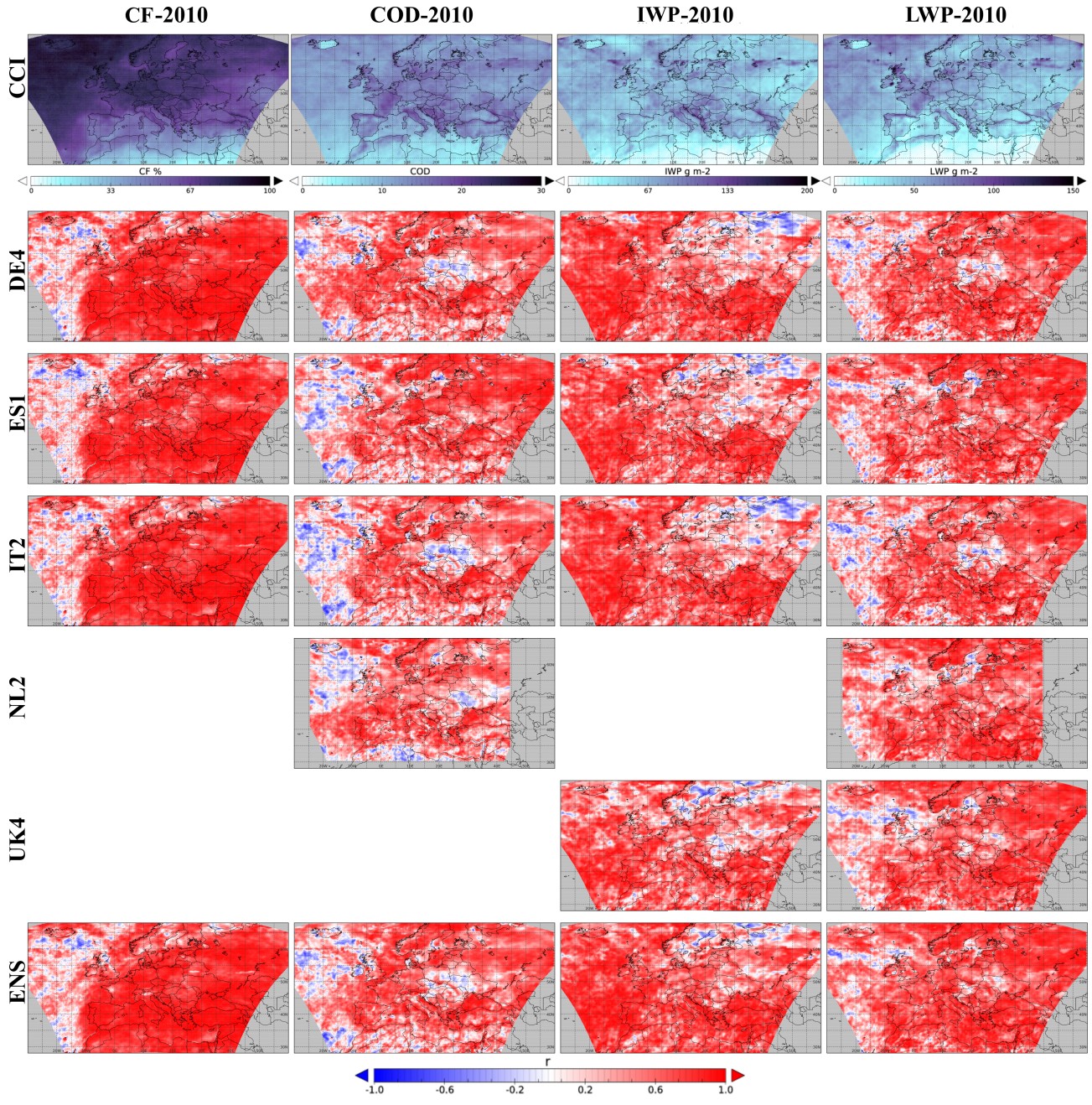

**Figure 2.** Temporal correlation for the whole year 2010. First row represents the mean satellite values of 2010 where each column represents a cloud variable. Following rows show the temporal correlation of each model and cloud variable.

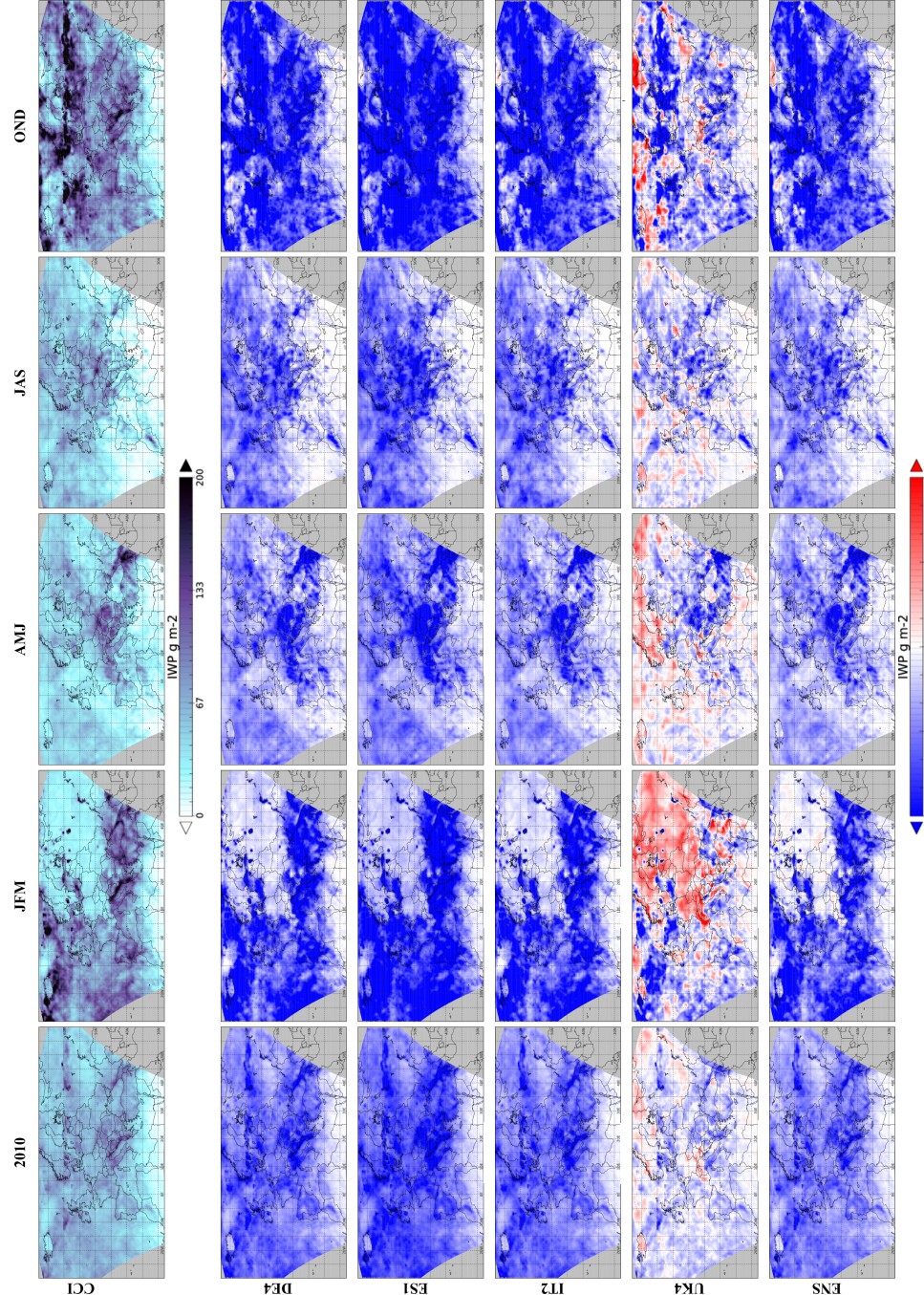

**Figure 3.** Same as Fig. 1 for the IWP and MEAN BIAS ERROR (bias).

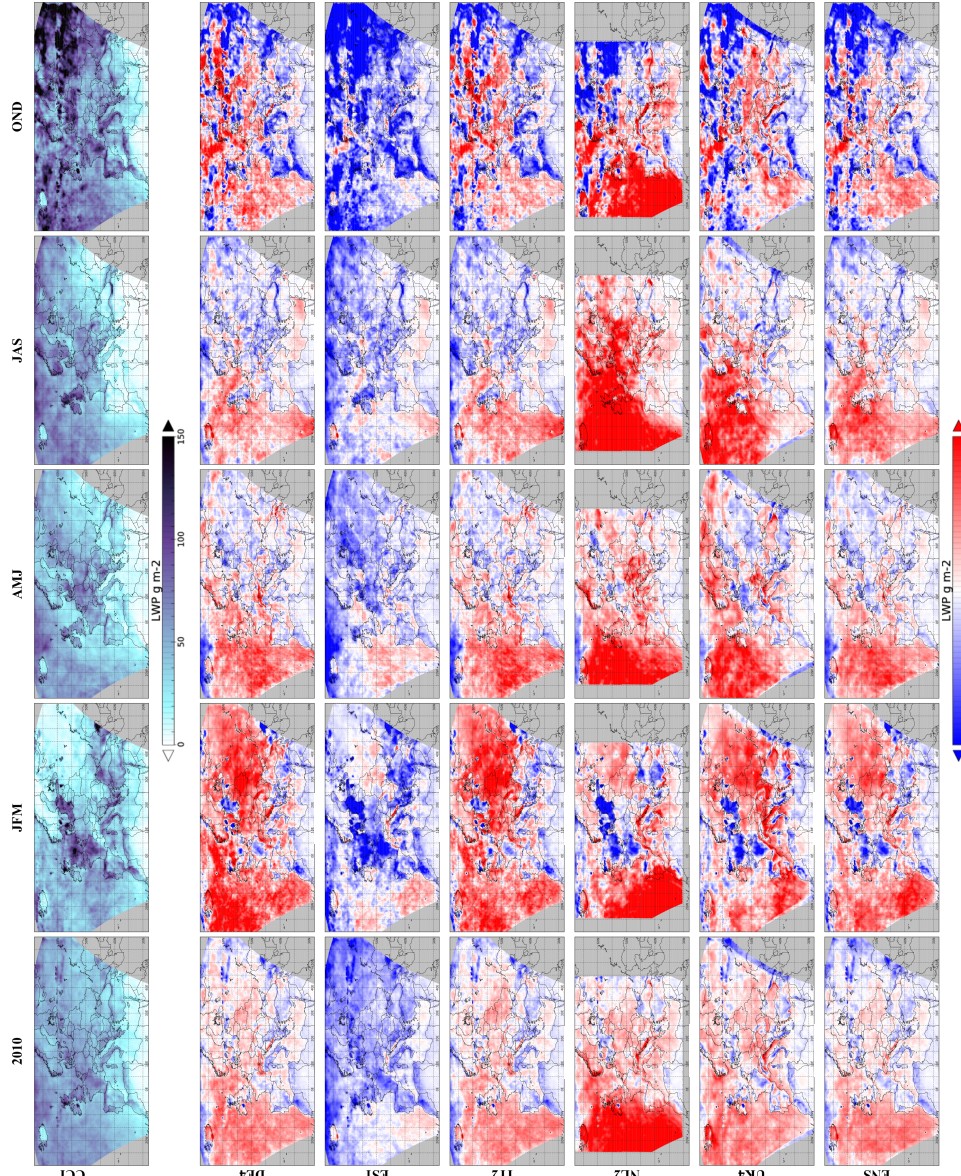

**Figure 4.** Same as Fig. 1 for the LWP and MEAN BIAS ERROR (bias).

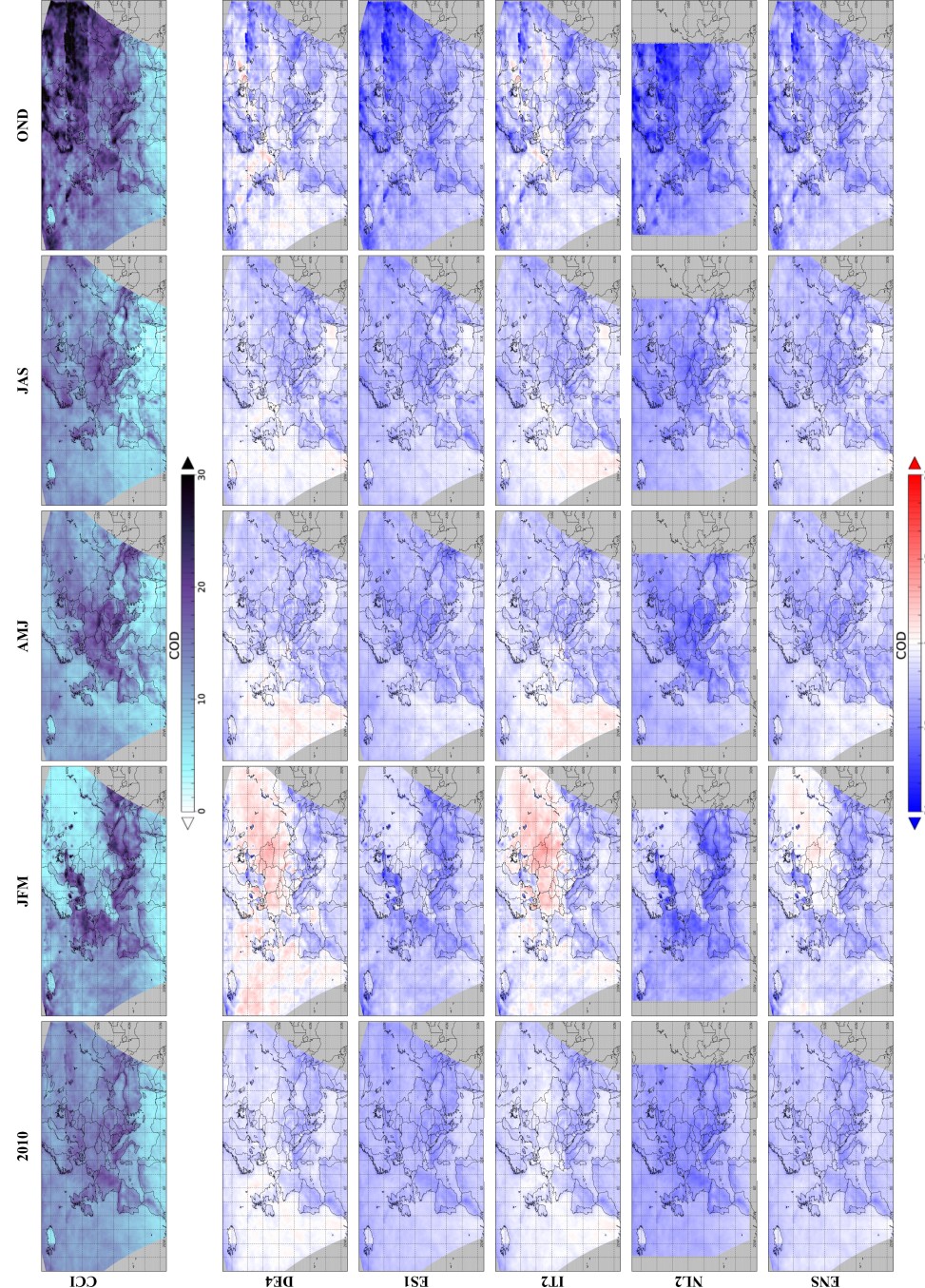

**Figure 5.** Same as Fig. 1 for the COD and MEAN BIAS ERROR (bias).

**Table 1.** Some of the AQMEII2 models features of the simulations studied.

| Model Simulation | Model | Microphysics | Gas Phase | SW radiation | LW radiation | Aerosol | Aerosol feedbacks |
|---|---|---|---|---|---|---|---|
| ES1 | | Lin | RADM2 | | | | |
| DE4 | WRF Chem | Morrison | RADM2 integ1 | RRTMG | RRTMG | MADE SORGAM | |
| IT2 | | | RACM | | | MADE VBS | Yes |
| NL2 | RACMO LOTOS-EUROS | Tiedke, Tompkins Neggers | CB-IV | RRTM | RRTM | ISORROPIA II 2 bins | |
| UK4 | METUM UKCA | Wilson & Ballard | RAQ | Edwards-Slingo | Edwards-Slingo | Classic | |

**Table 2.** Mean Satellite, models and Ensemble values for CF, COD, IWP and LWP.

| | CF MEAN VALUES | | | | | COD MEAN VALUES | | | | | IWP MEAN VALUES | | | | | LWP MEAN VALUES | | | | |
|---|---|---|---|---|---|---|---|---|---|---|---|---|---|---|---|---|---|---|---|---|
| | 2010 | JFM | AMJ | JAS | OND | 2010 | JFM | AMJ | JAS | OND | 2010 | JFM | AMJ | JAS | OND | 2010 | JFM | AMJ | JAS | OND |
| CCI | 62.5 | 70 | 60.8 | 53.3 | 67.2 | 11.4 | 10.3 | 11 | 10.5 | 14.5 | 52.2 | 63.3 | 42.2 | 36.2 | 74.8 | 42.3 | 36.3 | 36.3 | 39.6 | 59.8 |
| DE4 | 60.3 | 68.2 | 57.9 | 50.2 | 64.9 | 7 | 8.1 | 6.1 | 5.4 | 8.2 | 15 | 19.1 | 13 | 9.8 | 18 | 43.2 | 48 | 37 | 35.8 | 51.6 |
| ES1 | 53.6 | 60.3 | 50.6 | 46.3 | 57.1 | 3.2 | 3 | 3.2 | 3.2 | 3.5 | 7.1 | 9.2 | 6.4 | 4.7 | 8.2 | 28.1 | 24 | 25.2 | 27 | 29.4 |
| IT2 | 60.6 | 68.2 | 58.5 | 50.6 | 65 | 6.5 | 7.4 | 6 | 5.1 | 7.3 | 15 | 19 | 13.2 | 10 | 18 | 41.8 | 45.9 | 37.6 | 35.1 | 48.6 |
| NL2 | | | | | | 1.5 | 1.5 | 1.3 | 1.5 | 1.5 | | | | | | 58.8 | 53.4 | 52.2 | 64.5 | 65 |
| UK4 | | | | | | | | | | | 42.5 | 57 | 35.3 | 25.7 | 52 | 44.3 | 42.2 | 41.6 | 44.7 | 48.9 |
| ENS | 58.1 | 65.2 | 55.5 | 49.8 | 61.8 | 4.6 | 5.1 | 4.2 | 4 | 5.3 | 19.5 | 25.5 | 16.7 | 12.3 | 23.5 | 40.8 | 40.7 | 37 | 39 | 46.6 |

**Table 3.** Spatial correlation and standard deviation ratio values for CF, COD, IWP and LWP over the periods: 2010, JFM, AMJ, JAS, OND.
$r$: correlation coefficient; $\sigma_P/\sigma_O$: ratio between the standard deviation of the models ($\sigma_P$) and the observations ($\sigma_O$).

| | **CF** | | | | | | | | | |
|---|---|---|---|---|---|---|---|---|---|---|
| | **2010** | | **JFM** | | **AMJ** | | **JAS** | | **OND** | |
| | $r$ | $\sigma_P/\sigma_O$ | $r$ | $\sigma_P/\sigma_O$ | $r$ | $\sigma_P/\sigma_O$ | $r$ | $\sigma_P/\sigma_O$ | $r$ | |
| **DE4** | 0.89 | 1.06 | 0.94 | 1.01 | 0.94 | 1.02 | 0.94 | 1.07 | 0.95 | 1.06 |
| **ES1** | 0.87 | 1.05 | 0.89 | 1.03 | 0.91 | 1.05 | 0.92 | 1.03 | 0.92 | 1.06 |
| **IT2** | 0.88 | 1.07 | 0.94 | 1.03 | 0.94 | 1.03 | 0.94 | 1.07 | 0.95 | 1.07 |
| **NL2** | | | | | | | | | | |
| **UK4** | | | | | | | | | | |
| **ENS** | 0.85 | 1.06 | 0.93 | 1.04 | 0.92 | 1.04 | 0.93 | 1.05 | 0.94 | 1.07 |

| | **COD** | | | | | | | | | |
|---|---|---|---|---|---|---|---|---|---|---|
| | **2010** | | **JFM** | | **AMJ** | | **JAS** | | **OND** | |
| | $r$ | $\sigma_P/\sigma_O$ | $r$ | $\sigma_P/\sigma_O$ | $r$ | $\sigma_P/\sigma_O$ | $r$ | $\sigma_P/\sigma_O$ | $r$ | $\sigma_P/\sigma_O$ |
| **DE4** | 0.68 | 1.12 | 0.62 | 1.05 | 0.59 | 1.07 | 0.66 | 0.94 | 0.71 | 0.85 |
| **ES1** | 0.67 | 0.55 | 0.69 | 0.46 | 0.57 | 0.61 | 0.66 | 0.55 | 0.73 | 0.38 |
| **IT2** | 0.68 | 1.07 | 0.57 | 0.99 | 0.56 | 1.06 | 0.59 | 0.89 | 0.73 | 0.81 |
| **NL2** | 0.41 | 0.16 | 0.39 | 0.10 | 0.34 | 0.16 | 0.51 | 0.17 | 0.47 | 0.09 |
| **UK4** | | | | | | | | | | |
| **ENS** | 0.68 | 0.74 | 0.65 | 0.64 | 0.57 | 0.74 | 0.64 | 0.66 | 0.74 | 0.56 |

| | **IWP** | | | | | | | | | |
|---|---|---|---|---|---|---|---|---|---|---|
| | **2010** | | **JFM** | | **AMJ** | | **JAS** | | **OND** | |
| | $r$ | $\sigma_P/\sigma_O$ | $r$ | $\sigma_P/\sigma_O$ | $r$ | $\sigma_P/\sigma_O$ | $r$ | $\sigma_P/\sigma_O$ | $r$ | $\sigma_P/\sigma_O$ |
| **DE4** | 0.62 | 0.24 | 0.53 | 0.20 | 0.64 | 0.30 | 0.35 | 0.26 | 0.58 | 0.16 |
| **ES1** | 0.60 | 0.10 | 0.43 | 0.09 | 0.58 | 0.12 | 0.31 | 0.12 | 0.53 | 0.07 |
| **IT2** | 0.62 | 0.24 | 0.52 | 0.20 | 0.63 | 0.30 | 0.35 | 0.26 | 0.58 | 0.16 |
| **NL2** | | | | | | | | | | |
| **UK4** | 0.63 | 0.81 | 0.54 | 0.78 | 0.25 | 0.80 | 0.16 | 0.80 | 0.55 | 0.63 |
| **ENS** | 0.62 | 0.34 | 0.54 | 0.29 | 0.48 | 0.34 | 0.26 | 0.34 | 0.59 | 0.24 |

| | **LWP** | | | | | | | | | |
|---|---|---|---|---|---|---|---|---|---|---|
| | **2010** | | **JFM** | | **AMJ** | | **JAS** | | **OND** | |
| | $r$ | $\sigma_P/\sigma_O$ | $r$ | $\sigma_P/\sigma_O$ | $r$ | $\sigma_P/\sigma_O$ | $r$ | $\sigma_P/\sigma_O$ | $r$ | $\sigma_P/\sigma_O$ |
| **DE4** | 0.68 | 1.08 | 0.57 | 1.14 | 0.67 | 1.12 | 0.76 | 0.90 | 0.65 | 0.85 |
| **ES1** | 0.66 | 0.69 | 0.53 | 0.69 | 0.64 | 0.76 | 0.77 | 0.67 | 0.65 | 0.48 |
| **IT2** | 0.66 | 1.04 | 0.56 | 1.09 | 0.64 | 1.11 | 0.72 | 0.87 | 0.66 | 0.81 |
| **NL2** | 0.44 | 1.31 | 0.28 | 1.32 | 0.56 | 1.34 | 0.72 | 1.49 | 0.39 | 0.87 |
| **UK4** | 0.73 | 1.10 | 0.62 | 0.97 | 0.69 | 1.22 | 0.78 | 1.19 | 0.74 | 0.76 |
| **ENS** | 0.65 | 1.03 | 0.56 | 0.97 | 0.68 | 1.07 | 0.76 | 0.99 | 0.66 | 0.72 |