# Peer review of "Evaluating cloud properties in an ensemble of regional on-line coupled models against satellite observations"

_Atmospheric Chemistry and Physics, 2018_

## Referee Comment (RC1) · Anonymous Referee #1 · 12 Mar 2018

The paper presents the evaluation of regional meteorology-chemistry models taking part in the AQMEII experiment using observations from the ESA Cloud_cci project in order to specifically evaluate how the different models represent cloud variables (cloud fraction, cloud optical depth, cloud liquid water path, cloud ice path). The comparison provides with interesting results regarding the models' performances and their ability to correctly simulate cloud properties. However the paper has major limitations as described in detail below. Therefore, I would recommend publication only if major changes are made.

Major comments
1) The text is too long, descriptive and vague, it presents problems in tis structure and many times it does not flow well. The abstract needs to be re-written (see comment below). The introduction is too descriptive and the reader is getting lost in the mass of information that is not needed to be mentioned. Its paragpraph looses its content and purpose. The first paragraph of the introduction (page2, lines 1-20) is too vague and general. Then in the following parts in the introduction the authors describe in detail the references in other works. There is no need to to describe each reference's work but only what is relevant to the present study. Also the introduction misses the presentation of the contents of the paper that should be placed at the end of the introduction. In general I would suggest that the revision by a person with English as a first language would significantly improve the text quality.
2) The abstract needs also to be re-written. It is not clear through the abstract to the reader what the paper is about. The authors mention in the abstract and throughout the text the evaluation of aerosol-radiation and aerosol cloud interactions. In my opinion this is deriving through the comparison of cloud properties in the models that include ARI and ACI interactions, so the evaluation of ARI and ACI is an impact of the present study but not the principal work and for that reason it should not be presented as such. On the contrary the authors should clearly state in the abstract that they evaluate the representation of cloud properties (CF, CWP, CIP, COD).
3) In the methodology part, the use of man absolute error (Figures 2, 5, 7 ,9) adds no additional information in the evaluation study compared to the mean bias and should be omitted. A suggestion that the authors could test and see if they obtain interesting results is to use instead the statistical metric of mean absolute bias that is The authors could maybe instead use the statistical metric of fractional bias that is not affected by the magnitude of the value to be evaluated, as this problem is mentioned many times in the text.

Specific minor comments
1) page 1, line 2: "The effect of atmospheric aerosols.." please precise the effect where?
2) page 1, line 2: "The evaluated simulations run". Delete are
3) page 1, line 8 : I suggest that you use here and in all the text CF for cloud fraction (instead of CFR), as it is more commonly used in the literature.
4) page 1, line 8: An underestimation by what? You could add by the model.
5) page 1, lines 9-10: over the whole domain. Please precise which domain.
6) page 1, lines 11-14: Need to be re-writtem in better English.
7) page 2, lines 1-2: I suggest that you delete the sentence "The study … climate science" as it is very generic and vague.
8) page 2, line 27: I would suggest that you replace : "By using observations and modeling, Avey et al. (2007) combined cloud retrievals..."

9) page 2, line 31: Add in the beginning of the sentence "They found that where the transport model..."

10) page 3, lines 25-26: Replace with : "is demanded, both at global and regional scales". And then "Particularly, ACI is still considered one of the most..."

11) page 4, lines 1-2: No need to mention the exact content of the work of Brunner et al., (2015).

12) page 4, lines 6-11: This paragraph repeats things already written. Instead you could present the contents of the paper.

13) page 4, line 20: Replace with "covers the year 2010".

14) page 4, line 24: Replace the sentence "According to..' with "LOTOS-EUROS is a semi-column...".

15) page 4, line 31: No need for Table 2, as it is clearly described in the text.

16) page 5, line 10 : (and throughout the text) the word "data" should be treated as plural. The data are…

17) page 5, lines 13-14: Delete "of which" and rephrase the sentence.

18) page 5, line 18: Delete "Latitude/Longitude".

19) page 5, lines 20-24:  No need to refer to the exact bugs fixed in the dataset.

20) page 5, line 27: Replace with "For the valuation of cloud variables".

21) page 5, line 28: Replace with "we use several statistical metrics"

22) page 5, lines 30-31: Delete the last sentence. No need to add this information.

23) page 6, lines 15-19: The paragraph is not at all understood, it needs to be rephrased.

24) page 7, line 2: Add "All the figures of the present study have the.."

25) page 7, line 6: The sentence needs to be rephrased.

26) page 7, first paragraph: In the Section 3 it is useful to precise the limits of the study area.

27) page 7, line 15: 'to over -35%" Please precise where, for example over the study region.

28) page 7, lines 22-24: The sentence indicates that there is a problem in the dataset, so the data that are considered as a reference for the evaluation are not trustworthy? This creates an important rpoblem to the study and needs to be considered.

29) page 8, line 1: Add in the sentence "This spatial pattern is clear in the COD bias..."

30) page 8, lines 3-5: Give some precise numbers to justify the conclusions.

31) page 8, line 9 : You mention in the beginning of the sentence 'The authors.." Please precise which authors?

32) page 8, line 34: Replace with "whereas UK4 considers only one.

33) page 9, line 7: Replace with "Positive correlations are practically found in the entire..."

34) page 9, line 10: and throughout the text: you refer to the word levels and you mean the CWP values. Using the word levels is confusing as it is usually used for the atmospheric vertical levels in the literature.

35) page 9, line 11: Delete 'can be found"

36) page 9, line 20: Replace "the variable" with "the CWP in the model".

37) page 9, lines 23-24: Please rephrase the sentence.

38) page 10, lines 21-24: Give some numbers to be more precise.

39) page 10, line 26: "must be well represented". Precise where?

40) page 11,lines 7-9: The sentence needs to be rephrased.

41) page 11, line 29: You refer to mean satellite data. Please be more precise by declaring tha data of which variable.

42) page 12, lines 23-29: This paragraph presents to vague conclusions and does not merit a position in the study. Instead you could mention for example how the study offers to the scientific community by searching and applying model improvements in the direction highlighted in the study.

43) page 19, Figure 1: Replace with : "First row represents the mean satellite values of 2010 (first column), JFM (second column)…." Also precise in the title which models : DE4 second row, ES1 third column etc…

44) pages 20-27: The other figures titles should be like : Same as Fig. 1 for the COD. Etc..

45) page 30: Please precise in the legend of Table 3 the exact domain.

---

## Referee Comment (RC2) · Anonymous Referee #2 · 6 Apr 2018

This paper is an evaluation of six simulations over Europe against satellite observations. It includes three different models with one that has three different combinations of microphysics, radiation and aerosol parameterizations. The evaluated variables are cloud fraction, cloud optical depth, cloud liquid water path and cloud ice water path. The results are interesting since these quantities are an important source of uncertainties in atmospheric models. However, major revisions are needed to clarify the text, tables and figures; and additional information is needed to complete this evaluation, both for the observation and for the model analysis.

General comments:

- Figures are overwhelming: too many panels with poor color-scale choices. It results in too much information in each figure, such that it discourages the reader to look

into the details of the results. To reduce the number of panels, two seasons could be chosen together with the annual mean. Please remove the empty space for the missing simulations and explain it in the text. Remove the subtitles that cannot be read anyway. Choose color-scales (both for the observations and for the simulations) that highlight the important messages you want to show for each variables.

- The text is too long, particularly the introduction. The focus of the paper is not the ARI nor the ACI but the cloud variables that are evaluated. A short paragraph on the context of these simulations (within the AQMEII project) would be sufficient. The rest of the introduction should bring the reader rapidly to the main focus of the paper.

- The abstract should be re-written with the focus of the paper in mind (and a good revision for missing words, wrong terminology, appropriate use of English language).

- There are many basic English errors. The text should be reviewed by a native English-speaker.

- Moreover, there is room for improvements in the writting since many ideas and concepts are confusing. Good revisions would certainly improve the quality of the paper.

- With respect to the different variables evaluated, I would suggest to present CWP, CIP before COD since the two water content variables have a direct impact on the later variable (COD). Moreover, a cloud total water content may be of interest since some models may have a diagnostic separation between ice and liquid water content (even for the satellite retrieval?).

- Please clarify if CWP and CIP are evaluated for in-cloud values or not. If it is for grid point values (not in-cloud), how can you interpret the results for CWP, CIP and COD if a model does not provide CFR?

- For the conclusion, the last two paragraphs seem out of the subject. A more general conclusion is needed (and much shorter). What does it tell us about the use of these models to assess ARI or ACI? One idea per paragraph would help fluidity.

- It seems to be an assessment of models that will be used for aerosols studies, maybe a description of how microphysics schemes are connected to aerosols would be interesting. For example, does the aerosol concentrations are used as CCN? If so, does it have an impact on the evaluated variables?

Specific comments:

-The title is misleading since it suggests a focus on the chemistry/aerosols.

-Be careful of the name of the variables and their abbreviations. Maybe a use of more "traditional" abbreviations would ease the reading of the text. For example: cloud fraction (CF), cloud liquid water path (LWP), cloud ice water path (IWP).

-The Mean absolute error (MAE) does not add any useful information. Standard deviation or RSME would be more useful.

-Table 1 is really not clear: short lines are not aligned together. Instead, repeat the information if necessary and use some highlighting if you want to show the differences between the 3 WRF simulations. Table 3 could be put into a 4 panels figure. This summary of results is important but information is lost with all the numbers.

-Table 4 could also be put into a figure.

Introduction:

-page 3, Lines 30-35, clarify the context of this study, without putting too much details on all the working groups, etc.

-page 4, lines 6-10: this study is not evaluating any of the aerosols effects nor the aerosols-clouds effects. Please rephrase.

Methodology:

-page 4, lines 24-26: This information on a specific model is not at the right place (it is not the general idea of this paragraph). Should be put elsewhere.

-It would be a good place to highlight some differences between the models, for example with respect to their microphysics schemes, to the connection with the aerosols, and to the connection with the radiative transfer scheme.

Results:

-page 7: why not choose the usual seasons (DJF, JJA . . .)? It could be easier to compare with other studies.

-Figure 1: more information is needed for the CFR between 0 and 1, please change the color-scale.

-Page 7, line11: CFR values are generally higher than 0. Please comment the other values between 0 and 1.

-page 7, line 21: which negative bias?

-Page 7, lines 23-24: More information is needed on the satellite limits of detection, uncertainties, etc. Is this an hypothesis or is it stated in an article?

-The number of days available should be indicated somewhere in the graphics to indicate the validity of the statistics (for example in the corner of each panel).

-Page 8, line 8: either the number concentration or the mixing ratio (these are very different things)

-page 8, line 21: "others". Please, be more specific.

-Page 8, COD: An important question is how COD is calculated in each model? Are all the ice categories used in this calculations? Are the effective radii explicitly calculated? Or is it a parameterized function, or even a fixed value? Even if the microphysics provides such information, it is often not used in the radiation transfer scheme. Is the COD presented here is a representative value of what the radiative transfer scheme "sees" or does it comes out of the microphysics scheme? What are the hypotheses about effective radii in the satellite retrievals?

-Page 8, last paragraph for CIP: "some areas...". Please, be more specific.

-Page8, CIP: what is the satellite uncertainties related to ice water path retrievals? This must be high and certainly close to the model biases. If so, what can the authors conclude about that variable for the different simulations? This aspect would be interesting to be developed in the text.

-Page 8, CIP: When talking about hydrometeor categories, be specific in the way it is called. "number of hydrometeors" is confusing and is not the general accepted term.

-Page 9, lines 1-2: A more important question about the number of ice categories in the different microphysics scheme, is what categories are actually passed to the radiation and are part of the CIP (and COD) calculations in the different models? For example, snow, as a distinct category, is often ignored for radiation but may be part of the COD calculation. What are the hypothesis used in the satellite observations?

-Page 9, CWP: Comparison with other studies of satellite retrievals are a good step, but it does not translate into total cloud CWP... (studies presented here are for low and middle clouds only?!).

-Page 9, line 20: How is the diagnostic CWP from the UK4 model is better or worst than the other models? Please comment. It seems that this model is in the middle of the model performance for CWP. What about CIP for this model, is it diagnosed as well? It should be mentioned. What is the diagnostic? Is it only a function of temperature?

-For NL2 and UK4, since no CFR are provided, how can CWP and CIP biases can be interpreted? In other words, does a CWP or CIP bias include a CFR bias? This is important to be clarified.

Technical corrections: (many corrections on the formulation are not listed since they are too many)

Abstract:

[Figure]

-line1: "On-line coupled" . . .please rephrase

-line 9: "cloud liquid ice path" . . . cloud ice water path

-line 10: "CWP bias is broadly overestimated"... either keep : a large positive bias or CWP is broadly overestimated

-line 12: "capacity" . . . please rephrase

Introduction:

-line 25: "integrated meteorology-atmospheric chemistry" . . . rephrase

-line 25: "demanded" . . . not the right term

-line 27: problem with the citation and parenthesis

Methodology:

-page 4, line 16: "allow analyzing", please rephrase.

-page 4, line 19: use the same variable names and abbreviations as before (see the abstract)

-page 4, line 31: "According to...", please, do not refer to a table like this. This comment is valid for every table of the article.

-Page 5, line12: correct the citation parentheses.

-page 5, line 14: "we". Please rephrase.

-page 5, line 17: repetition of the line 14. Please rephrase.

-page 5 line 18: "monthly summary". Please rephrase or explain.

-page 5, line20: "BRDF" not defined

-page 5 line 31: last phrase to be rewritten.

-Page 6: equation 1: MBE not defined. Please include Pi, Oi in the equation.

-Page 6, lines 17-19: Please rephrase.

Results:

-page 7, line 26: "sea" please use always the same terms (ocean vs. sea)

-page 8, line 1, line 3: missing words

Please also note the supplement to this comment:
https://www.atmos-chem-phys-discuss.net/acp-2018-114/acp-2018-114-RC2-
supplement.pdf

---

## Author Comment (AC1) · 15 Jun 2018

First of all, we would like to thank the reviewers for the valuable comments. Please see below our point-by-point replies (reviewer's comments are displayed in black, our replies in blue fonts).

Response to **reviewer #1:**

Major comments

1) The text is too long, descriptive and vague, it presents problems in tis structure and many times it does not flow well. The abstract needs to be re-written (see comment below). The introduction is too descriptive and the reader is getting lost in the mass of information that is not needed to be mentioned. Its paragraph looses its content and purpose. The first paragraph of the introduction (page2, lines 1- 20) is too vague and general. Then in the following parts in the introduction the authors describe in detail the references in other works. There is no need to to describe each reference's work but only what is relevant to the present study. Also the introduction misses the presentation of the contents of the paper that should be placed at the end of the introduction. In general, I would suggest that the revision by a person with English as a first language would significantly improve the text quality.

2) The abstract needs also to be re-written. It is not clear through the abstract to the reader what the paper is about. The authors mention in the abstract and throughout the text the evaluation of aerosol- radiation and aerosol cloud interactions. In my opinion this is deriving through the comparison of cloud properties in the models that include ARI and ACI interactions, so the evaluation of ARI and ACI is an impact of the present study but not the principal work and for that reason it should not be presented as such. On the contrary the authors should clearly state in the abstract that they evaluate the representation of cloud properties (CF, CWP, CIP, COD).

Response to 1 and 2: The referees are right; abstract has been changed according to your suggestions (and specifying that we asses the representation of several cloud properties). More over, the manuscript has been revised by a native English-speaker.

3) In the methodology part, the use of man absolute error (Figures 2, 5, 7 ,9) adds no additional information in the evaluation study compared to the mean bias and should be omitted. A suggestion that the authors could test and see if they obtain interesting results is to use instead the statistical metric of mean absolute bias that is The authors could maybe instead use the statistical metric of fractional bias that is not affected by the magnitude of the

value to be evaluated, as this problem is mentioned many times in the text.

Response: we thank the referee's suggestion, but the metrics proposed add very limited extra information with respect to the statistical metrics already included in the manuscript.

Specific minor comments

1) page 1, line 2: "The effect of atmospheric aerosols.." please precise the effect where?

Response: it has been specified the meaning of this sentence.

2) page 1, line 2: "The evaluated simulations run". Delete are

Response: The sentence has been changed to are performed

3) page 1, line 8 : I suggest that you use here and in all the text CF for cloud fraction (instead of CFR), as it is more commonly used in the literature.

Response: the abbreviation "CF" has been changed as suggested.

4) page 1, line 8: An underestimation by what? You could add by the model.

Response: by the model has been added to the sentence

5) page 1, lines 9-10: over the whole domain. Please precise which domain.

Response: "European" has been added to specify the domain.

6) page 1, lines 11-14: Need to be re-written in better English.

Response: these lines has been re-written.

7) page 2, lines 1-2: I suggest that you delete the sentence "The study ... climate science" as it is very generic and vague.

Response: this sentence has been deleted as suggested.

8) page 2, line 27: I would suggest that you replace: "By using observations and modeling, Avey et al. (2007) combined cloud retrievals..."

Response: this sentence has been replaced as suggested.

9) page 2, line 31: Add in the beginning of the sentence "They found that where the transport model..."

Response: this sentence has been added.

10) page 3, lines 25-26: Replace with : "is demanded, both at global and regional scales". And then "Particularly, ACI is still considered one of the most..."

Response: this sentence has been replaced as suggested.

11) page 4, lines 1-2: No need to mention the exact content of the work of Brunner et al., (2015).

Response: since this study was performed under the umbrella of the AQMEII2 initiative, we specify the papers where the models' evaluation was done, so that the reader could check our model evaluation regarding the cited variables. We have added "*An extensive model evaluation of the simulations showed herein…*" to clarify it.

12) page 4, lines 6-11: This paragraph repeats things already written. Instead you could present the contents of the paper.

Response: this paragraph has been modified and we have written the contents of the manuscript as suggested.

13) page 4, line 20: Replace with "covers the year 2010".

Response: this sentence has been replaced as suggested.

14) page 4, line 24: Replace the sentence "According to..' with "LOTOS-EUROS is a semi-column...".

Response: this sentence has been replaced as suggested.

15) page 4, line 31: No need for Table 2, as it is clearly described in the text.

Response: the table has been removed from the manuscript.

16) page 5, line 10 : (and throughout the text) the word "data" should be treated as plural. The data are...

Response: this has been corrected throughout the text.

17) page 5, lines 13-14: Delete "of which" and rephrase the sentence.

Response: the sentence has been rephrased.

18) page 5, line 18: Delete "Latitude/Longitude".

Response: "Latitude/Longitude" has been deleted.

19) page 5, lines 20-24: No need to refer to the exact bugs fixed in the dataset.

Response: we refer to the bux's fixed since (as cited in the manuscript), they lead to a significant reduction of random and systematic uncertainties of the data, in particular for the optical properties cloud effective radius and cloud optical thickness as well as therefrom derived cloud liquid and ice water path.

20) page 5, line 27: Replace with "For the valuation of cloud variables".

Response: it has been replaced as suggested.

21) page 5, line 28: Replace with "we use several statistical metrics"

Response: the sentence has been rephrased.

22) page 5, lines 30-31: Delete the last sentence. No need to add this information.

Response: this sentence has been deleted.

23) page 6, lines 15-19: The paragraph is not at all understood, it needs to be rephrased.

Response: the sentence has been rephrased.

24) page 7, line 2: Add "All the figures of the present study have the.."

Response: the sentence has changed as suggested.

25) page 7, line 6: The sentence needs to be rephrased.

Response: the sentence has been rephrased.

26) page 7, first paragraph: In the Section 3 it is useful to precise the limits of the study area.

Response: the limits of the studied area has been defined in this paragraph.

27) page 7, line 15: 'to over -35%" Please precise where, for example over the study region.

Response: "Over the studied region" has been added.

28) page 7, lines 22-24: The sentence indicates that there is a problem in the dataset, so the data that are considered as a reference for the evaluation are not trustworthy? This creates an important problem to the study and needs to be considered.

Response: There is not a problem in the data set: optically, very thin clouds (about 10%) are missed in the observations. In spite of this, there is no cloud detection capability difference between land and sea. Thus the differences we see in the difference plots (obs-models) indicate way too low cloudiness over land in the models.

29) page 8, line 1: Add in the sentence "This spatial pattern is clear in the COD bias..."

Response: the sentence has been changed.

30) page 8, lines 3-5: Give some precise numbers to justify the conclusions.

Response: it has been changed as suggested.

31) page 8, line 9: You mention in the beginning of the sentence 'The authors..." Please precise which authors?

Response: "the authors" referred to Baró et al., 2016, the paragraph was divided in 2 but it was a typo. It has been corrected.

32) page 8, line 34: Replace with "whereas UK4 considers only one.

Response: the sentence has been changed as suggested.

33) page 9, line 7: Replace with "Positive correlations are practically found in the entire..."

Response: the sentence has been changed as suggested.

34) page 9, line 10: and throughout the text: you refer to the word levels and you mean the CWP values. Using the word levels is confusing as it is usually used for the atmospheric vertical levels in the literature.

Response: "levels" have been changed to "values" in the context that the reviewer suggested.

35) page 9, line 11: Delete 'can be found"

Response: "can be found" has been deleted.

36) page 9, line 20: Replace "the variable" with "the CWP in the model".

Response: the sentence has been changed as suggested.

37) page 9, lines 23-24: Please rephrase the sentence.

Response: the sentence has been rephrased.

38) page 10, lines 21-24: Give some numbers to be more precise.

Response: more values have been added to the sentence.

39) page 10, line 26: "must be well represented". Precise where?

Response: the sentence has been changed.

40) page 11, lines 7-9: The sentence needs to be rephrased.

Response: the sentence has been rephrased.

41) page 11, line 29: You refer to mean satellite data. Please be more precise by declaring the data of which variable.

Response: LWP has been added since we referred to this mean satellite data.

42) page 12, lines 23-29: This paragraph presents to vague conclusions and does not merit a position in the study. Instead you could mention for example how the study offers to the scientific community by searching and applying model improvements in the direction highlighted in the study.

Response: This paragraph has been modified and we have followed the referee's suggestions.

43) page 19, Figure 1: Replace with : "First row represents the mean satellite values of 2010 (first column), JFM (second column)...." Also precise in the title which models : DE4 second row, ES1 third column etc... 44) pages 20-27: The other figures titles should be like : Same as Fig. 1 for the COD. Etc..

Response: the figures capture has been changed as suggested.

45) page 30: Please precise in the legend of Table 3 the exact domain.

Response: the domain has been specified.

Response to **reviewer #2**

General comments:

• Figures are overwhelming: too many panels with poor color-scale choices. It results in too much information in each figure, such that it discourages the reader to look into the details of the results. To reduce the number of panels, two seasons could be chosen together with the annual mean. Please remove the empty space for the missing simulations and explain it in the text. Remove the subtitles that cannot be read anyway. Choose color-scales (both for the observations and for the simulations) that highlight the important messages you want to show for each variables.

Response: The figures have been redone, removing the white empty spaces and the subtitles. Also, the color bar of the satellite mean values has been changed to see better the values between 0 and 1. We considered to leave all the seasons since some significant results appears in the different seasons for the different variables.

• The text is too long, particularly the introduction. The focus of the paper is not the ARI nor the ACI but the cloud variables that are evaluated. A short paragraph on the context of these simulations (within the AQMEII project) would be sufficient. The rest of the introduction should bring the reader rapidly to the main focus of the paper.
• The abstract should be re-written with the focus of the paper in mind (and a good revision for missing words, wrong terminology, appropriate use of English language).
• There are many basic English errors. The text should be reviewed by a native English-speaker. Moreover, there is room for improvements in the writing since many ideas and concepts are confusing. Good revisions would certainly improve the quality of the paper.

Response: The referees are right; abstract has been changed according to your suggestions (and specifying that we asses the representation of several cloud properties). And the manuscript has been revised by a native English-speaker.

• With respect to the different variables evaluated, I would suggest to present CWP, CIP before COD since the two water content variables have a direct impact on the later variable (COD). Moreover, a cloud total water content may be of interest since some models may have a diagnostic separation between ice and liquid water content (even for the satellite retrieval?).

Response: The order of the variables has been changed as suggested. Regarding the suggestion of the cloud total water content variable, unfortunately, it was not included within the output variables of the AQMEII initiative.

• Please clarify if CWP and CIP are evaluated for in-cloud values or not. If it is for grid point values (not in-cloud), how can you interpret the results for CWP, CIP and COD if a model does not provide CFR?

Response: They all-sky mean was computed and this information has been added to the manuscript.

• For the conclusion, the last two paragraphs seem out of the subject. A more general conclusion is needed (and much shorter). What does it tell us about the use of these models to assess ARI or ACI? One idea per paragraph would help fluidity.

Response: These paragraphs has been changed.

• It seems to be an assessment of models that will be used for aerosols studies, maybe a description of how microphysics schemes are connected to aerosols would be interesting. For example, does the aerosol concentrations are used as CCN? If so, does it have an impact on the evaluated variables?

Response: when using offline modeling, there is a fixed aerosol concentration whereas in online-coupled models, this aerosol concentration is changing along time. So, in this sense, it will have an impact, (which is supposed to be more similar to reality, since in there is no aerosol conetration fixed, and with this study, we wanted to asses how these online models represent cloud variables).

**Specific comments:**

• The title is misleading since it suggests a focus on the chemistry/aerosols.

Response: The title has been change to "Evaluating cloud properties in an ensemble of regional on-line coupled models against satellite observations"

• Be careful of the name of the variables and their abbreviations. Maybe a use of more "traditional" abbreviations would ease the reading of the text. For example: cloud fraction (CF), cloud liquid water path (LWP), cloud ice water path (IWP).

Response: the name of the variables and their abbreviations has been changed as suggested

• The Mean absolute error (MAE) does not add any useful information. Standard deviation or RSME would be more useful.

Response: we thank the referee's suggestion, but the metrics proposed add very limited extra information with respect to the statistical metrics already included in the manuscript.

• Table 1 is really not clear: short lines are not aligned together. Instead, repeat the information if necessary and use some highlighting if you want to show the differences between the 3 WRF simulations.

Response: Some of the names has been better aligned since they were not. We tried different displays but we found them more confusing than the actual table. With the better alignment we considered to keep as it is.

• Table 3 could be put into a 4 panels figure. This summary of results is important but information is lost with all the numbers. Table 4 could also be put into a figure.

Response: We tried different combinations and we considered to leave it as it was. Moreover, there are already a lot of plots.

• Introduction:
◦ page 3, Lines 30-35, clarify the context of this study, without putting too much details on all the working groups, etc.

Response: this paragraph has been reduced and the context clarified.

◦ page 4, lines 6-10: this study is not evaluating any of the aerosols effects nor the aerosols-clouds effects. Please rephrase.

Response: this sentence has been rephrased.

• Methodology:
◦ page 4, lines 24-26: This information on a specific model is not at the right place (it is not the general idea of this paragraph). Should be put elsewhere.

Response: We have specified that the simulations were conducted by online coupled models so that the information of the model LOTOS fits better in this paragraph.

◦ It would be a good place to highlight some differences between the models, for example with respect to their microphysics schemes, to the connection with the aerosols, and to the connection with the radiative transfer scheme.

Response: we thank the referee's suggestion, to avoid repetition (since it is described in the resutls) and ending up with a long paragraph, we considered no to describe it.

• Results:
◦ page 7: why not choose the usual seasons (DJF, JJA ...)? It could be easier to compare with other studies.

Response: The referee is right but in this case, we chose this groups of months since we only have the year 2010, so we didn't have information of the previous December 2009.

◦ Figure 1: more information is needed for the CFR between 0 and 1, please change the color- scale.

Response: We have changed the color-bar of the mean values to that it can be

◦ Page 7, line11: CFR values are generally higher than 0. Please comment the other values between 0 and 1.

Response: this has been commented in the text.

◦ page 7, line 21: which negative bias?

Response: There was a typo, instead of "this" should has been "the".

◦ Page 7, lines 23-24: More information is needed on the satellite limits of detection, uncertainties, etc. Is this an hypothesis or is it stated in an article? The number of days available should be indicated somewhere in the graphics to indicate the validity of the statistics (for example in the corner of each panel).

Response: optically very thin clouds (about 10% of the clouds) are missed in the observations. In spite of this, there is no cloud detection capability difference between land and sea. Thus the differences we see in the difference plots (obs-models) indicate way too low cloudiness over land in the models.

◦ Page 8, line 8: either the number concentration or the mixing ratio (these are very different things)

Response: In this line, we refer to the WRF variable "QNDROP" which is defined as the "cloud droplet number mixing ratio".

∘ page 8, line 21: "others". Please, be more specific.

Response: "Others" here is a typo, the sentence has been changed.

∘ Page 8, COD: An important question is how COD is calculated in each model? Are all the ice categories used in this calculations? Are the effective radii explicitly calculated? Or is it a parameterized function, or even a fixed value? Even if the microphysics provides such information, it is often not used in the radiation transfer scheme. Is the COD presented here a representative value of what the radiative transfer scheme "sees" or does it comes out of the microphysics scheme? What are the hypotheses about effective radii in the satellite retrievals?

Response: the COD presented here comes out of the microphysics scheme. We have added some information of the hypotheses about effective radii in the satellite retrievals to the observations description in section 2.2.

∘ Page 8, last paragraph for CIP: "some areas...". Please, be more specific.

Response: I am sorry; I haven't found "some areas..." in this paragraph.

∘ Page 8, CIP: what is the satellite uncertainties related to ice water path retrievals? This must be high and certainly close to the model biases. If so, what can the authors conclude about that variable for the different simulations? This aspect would be interesting to be developed in the text.

Response: the satellite uncertainties related to ice water path retrievals has been developed in the manuscript, at the end of Section 2.2.

∘ Page 8, CIP: When talking about hydrometeor categories, be specific in the way it is called. "number of hydrometeors" is confusing and is not the general accepted term.

Response: we have added "species" after hydrometeors,

∘ Page 9, lines 1-2: A more important question about the number of ice categories in the different microphysics scheme, is what categories are actually passed to the radiation and are part of the CIP (and COD) calculations in the different models? For example, snow, as a distinct category, is often ignored for radiation but may be part of the COD calculation. What are the hypothesis used in the satellite observations?

Response: we have added some information related to the the hypothesis used in the satellite observations on the observational approach to the observations description in section 2.2.

∘ Page 9, CWP: Comparison with other studies of satellite retrievals are a good step, but it does not translate into total cloud CWP... (studies presented here are for low and middle clouds only?!).

Response: the referee is right, we have changed it to the study of Greenwald et al., 1993, in which they retrieve integrated LWP.

∘ Page 9, line 20: How is the diagnostic CWP from the UK4 model is better or worst than the other models? Please comment. It seems that this model is in the middle of the model performance for CWP. What about CIP for this model, is it diagnosed as well? It should be mentioned. What is the diagnostic? Is it only a function of temperature?

Response: The referee is right, UK4 model shows similar results to the ensemble mean, being in the middle of the model. For this model, CIP is a prognostic variable. This has been added to the manuscript.

∘ For NL2 and UK4, since no CFR are provided, how can CWP and CIP biases can be interpreted? In other words, does a CWP or CIP bias include a CFR bias? This is important to be clarified.

Response: CWP or CIP doesn't include CFR bias. Models NL2 and UK4 didn't provide CFR as an output variable within the AQMEII2 initiative, but these models had also cloud information that was taken to compute the CWP or CIP.

**Technical corrections**: (many corrections on the formulation are not listed since they are too many)
• Abstract,
∘ line1: "On-line coupled" ...please rephrase
Response: rephrased.

∘ line 9: "cloud liquid ice path" ... cloud ice water path
Response: corrected

∘ line 10: "CWP bias is broadly overestimated"... either keep : a large positive bias or CWP is broadly overestimated
Response: corrected

◦ line 12: "capacity" ... please rephrase
Response: rephrased.

• Introduction:
◦ line 25: "integrated meteorology-atmospheric chemistry" ... rephrase

Response: this sentence has been rephrased.

◦ line 25: "demanded" ... not the right term

Response: "demanded" has been changed for "needed"

◦ line 27: problem with the citation and parenthesis
Response: corrected

• Methodology:
◦ page 4, line 16: "allow analysing", please rephrase.
Response: this sentence has been rephrased.

◦ page 4, line 19: use the same variable names and abbreviations as before (see the abstract)
Response: corrected

◦ page 4, line 31: "According to...", please, do not refer to a table like this. This comment is valid for every table of the article.
Response: corrected

◦ Page 5, line12: correct the citation parentheses.
Response: corrected

◦ page 5, line 14: "we". Please rephrase.
Response: rephrased.

◦ page 5, line 17: repetition of the line 14. Please rephrase.
Response: rephrased.

◦ page 5 line 18: "monthly summary". Please rephrase or explain.
Response: It is related to monthly means values. This has been changed in the text.

◦ page 5, line20: "BRDF" not defined
Response: BRDF has been defined.

◦ page 5 line 31: last phrase to be rewritten.
Response: rewritten.

◦ Page 6: equation 1: MBE not defined. Please include Pi, Oi in the equation.
Response: changed.

◦ Page 6, lines 17-19: Please rephrase.
Response: rephrased.

• Results:

◦ page 7, line 26: "sea" please use always the same terms (ocean vs. sea)
Response: this has been unified.

◦ page 8, line 1, line 3: missing words
Response: The Word "in" was missing and added.

---

## Referee Report (RR1)

Evaluating cloud properties in an ensemble of regional on-line coupled models against satellite
observations
R. Baro et al.

This paper is an evaluation of six simulations over Europe against satellite observations. It includes three different models with one that has three different combinations of microphysics, radiation and aerosol parameterizations. The evaluated variables are cloud fraction, cloud optical depth, cloud liquid water path and cloud ice water path. The results are interesting since these quantities are an important source of uncertainties in atmospheric models. However, major revisions are needed to clarify the text, tables and figures; and additional information is needed to complete this evaluation, both for the observation and for the model analysis.

This is the 2$^{nd}$ revision.

**General comments:**
- A significant amount of work has been done for this round of corrections and it clarifies notably the text and the ideas. However, some work is still needed in the analysis, particularly when linking biases to effective radius.
- Again, as mentioned previously, Mean Absolute Error does not add any information for the analysis. Whenever it is mentioned in the text, it is to emphasize that the values have the same maxima as the bias. Moreover, many figures of MAE have scales from -50/-40 to +40/+50... how can it be negative based on equation 2? Figure 7 is not even mentioned in the text. Removal of these figures should be considered seriously even if no other metrics is proposed as an alternative.
- Figures should be following the order that they are mentioned in the text.
- In the satellite data, a strange line or a discontinuity appears around 60N (with very high values in very small locations, particularly for JFM, OND). Is there some problem with the retrieval around this area? If so, it should be mentioned in the text.
- 3.2 the fact that the UK4 simulation has the smaller IWP biases but only has one ice category is exactly the reverse of the argument there. And there is no mention about the other models, how many ice category do they have?
- 3.3 lines 15 and after: this analysis is not clear at all. The reference to Tiedke does not seems to fit in the argument. Why is the data not available over Northern Africa?
- If LWP and IWP are " all-sky means", these variables biases will include the CF biases. This could be the main differences between the simulations. One has to be very careful about over interpreting the LWP, IWP and COD biases when it includes the CF biases. That is the main weakness of the article.
- The links between effective radius and biases seen in LWP is weak if not even erroneous. The impact of effective radius will be on COD, but the author has to keep in mind that contributions from CF, LWP and IWP are also important to the COD biases.

**Specific comments:**
- The authors mention that they removed subtitle from the figures but it is still there and still too small to be read. This should be removed.
- Introduction, line 31: remove "they did not find any conclusive evidence"
- Introduction, last phrase: rephrase to removed "we"
- 2.2 last paragraph: is the -114 g/m2 bias is the mean bias against DARDAR? If it is, it can

change a lot of the interpretation of that variable. More details about these biases should be provided.

- 3.1, line24: do you mean between 0 and 1 %? please, clarify.
- 3.1, line 4 and 5: not coherent bewteen -35% and -40%
- Please remove figure 2, as the text explain it well, it coincides with the bias!
- 3.1, line8-10, please rephrase since it seems that the explanation is redundant.
- 3.1, last phrase: the highest correlation is over land... what is the argument there?
- 3.2 please put the figures in order that they appear in text.
- 3.3 : rephrase line 10-12 and clarify the general idea.
- Conclusion: line 28 and line 30: the reverse is stated between the two sentences. The argument for the IWP and the number of ice categories does not hold.
- Conclusion: lines 6-9: the argument that a smaller effective radius is responsible for a higher LWP does not hold. This whole argument should be kept to the interpretation of COD biases, but keep in mind that contributions from CF, LWP and IWP are important to the COD biases (lines 14-22).
- Conclusion line 30: Where is the conclusion about Morrison's scheme comes from? The IWP biases do not show that.
- Conclusion, last phrase: "we help to show". Please rephrase.

**Technical points:**
- 2.2, line 18: remove " utilised"
- 2.2, line 19: unclear: "which forms part"
- 2.3, line 13: "valuation", line 14: werw
- 2.3, last phrase: re-phrase
- 3, line 11: "provides" should be replace
- 3, line 12: removed "are presented"
- 3, last phrase: removed
- 3.1, line 7: "a nhe" ?
- 3.2 line 29: "progrosticked", do you mean diagnostic or prognostic?
- 3.2 ice hydrometeor categories
- 3.3 line 13: removed " included"
- 3.4, line 26: "vales"
- 3.5, line 28-29: rephrase
- 3.5 line 2: rephrase "pervasively"
- conclusion, line 31: "MAjor"

---

## Author Response (AR2)

We would like to thank once again the reviewers for the valuable comments and the time spent. Please see below our point-by-point replies (reviewer's comments are displayed in black, our replies in blue fonts). *We have also taken into account the suggestions for revision of the **Reviewer #1**.*

General comments:

• A significant amount of work has been done for this round of corrections and it clarifies notably the text and the ideas. However, some work is still needed in the analysis, particularly when linking biases to effective radius.

• Again, as mentioned previously, Mean Absolute Error does not add any information for the analysis. Whenever it is mentioned in the text, it is to emphasize that the values have the same maxima as the bias. Moreover, many figures of MAE have scales from -50/-40 to +40/+50... how can it be negative based on equation 2? Figure 7 is not even mentioned in the text. Removal of these figures should be considered seriously even if no other metrics is proposed as an alternative.

Response: As suggested also by the Reviewer#1, MAE has been removed from the manuscript.

• Figures should be following the order that they are mentioned in the text.

Response: The reviewer is right, now they are cited in order.

• In the satellite data, a strange line or a discontinuity appears around 60N (with very high values in very small locations, particularly for JFM, OND). Is there some problem with the retrieval around this area? If so, it should be mentioned in the text.

Response: Yes. This feature is very likely cause by the fact that for December regions north of 60N do not have daytime observations, due to illuminations conditions, but only night time observations instead, and that some (minor) retrieval inconsistency between day, twilight and night cause the feature seen. We have added a sentence in the text to clarify it.

• 3.2 the fact that the UK4 simulation has the smaller IWP biases but only has one ice category is exactly the reverse of the argument there. And there is no mention about the other models, how many ice categories do they have?

Response: The reviewer is right; the argument was reverse written.

• 3.3 lines 15 and after: this analysis is not clear at all. The reference to Tiedke does not seems to fit in the argument. Why is the data not available over Northern Africa?

Response: We considered to leave the argument of Tiedke here because it refers to the representation of this variable. Values over North Africa are very low, specifically during summer months, where there are no LWP. This is due to the proximity to the desert, and even if there is some LWP, as it is a monthly mean, it is close to 0.

• If LWP and IWP are "all-sky means", these variables biases will include the CF biases. This could be the main differences between the simulations. One has to be very careful about over interpreting the LWP, IWP and COD biases when it includes the CF biases. That is the main weakness of the article.

• The links between effective radius and biases seen in LWP is weak if not even erroneous. The impact of effective radius will be on COD, but the author has to keep in mind that contributions from CF, LWP and IWP are also important to the COD biases.

Response: The reviewer is right and we are aware of that weakness. We used the all-sky means since that's how all the available model output was provided. According to Greenwald et al., 2018, the "all-sky" CLWP is a more relevant variable for comparison. Others studies (Karlsson et al, 2017) also use all-sky values.

Greenwald, T. J, et al. (2018). An Uncertainty Data Set for Passive Microwave Satellite Observations of Warm Cloud Liquid Water Path. Journal of Geophysical Research: Atmospheres, vol. 123, 3668–3687. https://doi.org/10.1002/2017JD027638

Karlsson, K-G, et al. (2017). CLARA-A2: the second edition of the CM SAF cloud and radiation data record from 34 years of global AVHRR data. Atmospheric Chemistry and Physics, vol. 17, no 9, 5809-5828.

Specific comments:

• The authors mention that they removed subtitle from the figures but it is still there and still too small to be read. This should be removed.
Response: The reviewer is right, now the small titles are removed.

• Introduction, line 31: remove "they did not find any conclusive evidence"
Response: This sentence has been removed.

• Introduction, last phrase: rephrase to removed "we"
Response: "we" has been removed.

• 2.2 last paragraph: is the -114 $g/m^2$ bias is the mean bias against DARDAR? If it is, it can change a lot of the interpretation of that variable. More details about these biases should be provided.
Response: Yes, the global mean bias against the best available reference (DARDAR) is -114 $g/m^2$. The models show even lower IWP values, and that this, together with the negative CCI bias against DARDAR, just strengthens the conclusion that models have way too small IWP. This has been added to the manuscript.

• 3.1, line24: do you mean between 0 and 1 %? please, clarify.
Response: This has been clarified in the manuscript.

• 3.1, line 4 and 5: not coherent between -35% and -40%
Response: This sentences has been removed from the manuscript since we removed the MAE.

• Please remove figure 2, as the text explain it well, it coincides with the bias!
Response: MAE has been removed from the manuscript.

• 3.1, line8-10, please rephrase since it seems that the explanation is redundant.
Response: This lines has been rephrased.

• 3.1, last phrase: the highest correlation is over land... what is the argument there?

Response: As stated in the manuscript, since satellite missed thin clouds over sea areas, we see higher correlation over land.

• 3.2 please put the figures in order that they appear in text.

Response: This has been corrected.

• 3.3: rephrase line 10-12 and clarify the general idea.

Response: These lines has been rephrased.

• Conclusion: line 28 and line 30: the reverse is stated between the two sentences. The argument for the IWP and the number of ice categories does not hold.

Response: The reviewer is right; this sentence has been corrected.

• Conclusion: lines 6-9: the argument that a smaller effective radius is responsible for a higher LWP does not hold. This whole argument should be kept to the interpretation of COD biases, but keep in mind that contributions from CF, LWP and IWP are important to the COD biases (lines 14-22).

Response: According to lines 6-9, we suggested that the higher values of LWP could be related to that, but not that this is the cause. We tried to explained that since in the study of Baró et al, they compared the same model simulation, differing only in the microphysics (Lin or Morrison). We have seen a different behaviour between this two microphysics schemes.

• Conclusion line 30: Where is the conclusion about Morrison's scheme comes from? The IWP biases do not show that.

Response: The reviewer is right; this sentence has been removed.

• Conclusion, last phrase: "we help to show". Please rephrase.

Response: the sentence has been rephrased.

All the following technical points has been corrected:

Technical points:
• 2.2, line 18: remove " utilised"
• 2.2, line 19: unclear: "which forms part"
• 2.3, line 13: "valuation", line 14: werw
• 2.3, last phrase: re-phrase
• 3, line 11: "provides" should be replace
• 3, line 12: removed "are presented"
• 3, last phrase: removed
• 3.1, line 7: "a nhe" ?
• 3.2 line 29: "progrosticked", do you mean diagnostic or prognostic?
• 3.2 ice hydrometeor categories
• 3.3 line 13: removed " included"
• 3.4, line 26: "vales"
• 3.5, line 28-29: rephrase
• 3.5 line 2: rephrase "pervasively"
• conclusion, line 31: "MAjor"

[revised manuscript text omitted]